# BED-LLM: Intelligent Information Gathering with LLMs and Bayesian Experimental Design

**Deepro Choudhury**[*]   **Sinead Williamson**[†]   **Adam Goliński**[†]   **Ning Miao**[‡]
**Freddie Bickford Smith**[*]   **Michael Kirchhof**[†]   **Yizhe Zhang**[†]   **Tom Rainforth**[*]

[*]University of Oxford    [†]Apple    [‡]City University of Hong Kong

## Abstract

We propose a general-purpose approach for improving the ability of large language models (LLMs) to intelligently and adaptively gather information from a user or other external source using the framework of sequential Bayesian experimental design (BED). This enables LLMs to act as effective multi-turn conversational agents and interactively interface with external environments. Our approach, which we call BED-LLM (Bayesian experimental design with large language models), is based on iteratively choosing questions or queries that maximize the expected information gain (EIG) with respect to a variable of interest given the responses gathered previously. We show how this EIG can be formulated (and then estimated) in a principled way using a probabilistic model derived from the LLM's predictive distributions and provide detailed insights into key decisions in its construction and updating procedure. We find that BED-LLM achieves substantial gains in performance across a wide range of tests based on the 20 Questions game and using the LLM to actively infer user preferences, compared to purely prompting-based design generation and other adaptive design strategies.

## 1 Introduction

Intelligent information gathering—the ability to ask the right questions at the right time—is fundamental to effective AI systems. However, despite their many successes, LLMs currently fall short on proactively seeking out information from a user or external environment in an adaptive manner (Laban et al., 2025; Li et al., 2025c). For example, they have been shown to perform poorly on multi-turn guessing games (Bertolazzi et al., 2023; Zhang et al., 2024), task clarification (Chi et al., 2024), IT task automation (Jha et al., 2025) and multi-step tool use (Patil et al., 2025). In particular, while modern LLMs are often capable of producing coherent and insightful questions (or other external queries) in a single-turn setting, they typically struggle to appropriately tailor their questions to previously gathered responses on interactive tasks (Bertolazzi et al., 2023; Patil et al., 2025).

There is, therefore, a pressing need to improve the ability of LLMs to *adaptively* ask questions based on previous responses, and gather information in a targeted manner. Such capabilities are essential for a wide variety of problems, such as clarifying user intent, personalizing model behavior to a particular user, or generally acting as effective multi-turn conversational agents. They are also critical if we want to use LLMs in data gathering tasks or as automated agents in decision-making pipelines (Wu et al., 2025). In turn, these capabilities are essential across domains ranging from medical diagnosis (Hirosawa et al., 2024), troubleshooting (Jha et al., 2025), preference learning (Chakraborty et al., 2024; Handa et al., 2024; Ouyang et al., 2022) and tutoring systems (Kestin et al., 2025; Liu et al., 2024a), to conducting automated surveys (Aher et al., 2023; Jacobsen et al., 2025; Lee et al., 2024) and AI-driven scientific inquiries (Lu et al., 2024; Mandal et al., 2025). Note that in all these problems it is not enough for the LLM to generate full sets of suitable questions up front: we need it to be able to adaptively choose questions that are tailored to the already-collected user responses.

We propose to address this challenge using the framework of *sequential Bayesian experimental design* (BED; Chaloner & Verdinelli, 1995; Lindley, 1956; MacKay, 1992; Rainforth et al., 2024; Sebastiani & Wynn, 2000), which provides a model-based, information-theoretic mechanism for making adaptive design decisions, given a generative model of the experiment. Specifically, we show how the problem of interactive information gathering with LLMs can be formulated as a

sequential experimental design problem with a model derived from the LLM, wherein we iterate between choosing queries by maximizing expected information gain (EIG) in a variable of interest and updating our beliefs with the information from the received response.

We call our approach BED-LLM and show how its success is critically dependent on our precise model formulation, belief updating procedure and EIG estimation strategy. In particular, we show that it is essential to formulate the model with a precise distribution pairing that does not solely rely on in-context learning to update beliefs and uses the LLM's uncertainties in the space of answers rather than the more complicated underlying hypothesis space we are trying to learn in.

Together, we find that these innovations provide substantial performance benefits over directly generating queries from the LLM and more basic approximations of the sequential BED framework. Specifically, we first find that BED-LLM provides substantial improvements in the success rate for the 20 Questions problem across a variety of LLMs and target quantities. On average, BED-LLM improves the success rate by 37.4 percentage points compared to direct prompting of the LLM, with the success rate more than doubling in over half of the setups considered and never decreasing. Second, we demonstrate noticeable improvements in using the LLM for movie recommendations, showing that these benefits hold even when the LLM's predictive model differs from that of the answerer.

## 2 PROBLEM FORMULATION AND BACKGROUND

There are two natural ways to improve LLMs' ability to gather information: modifying the model itself (e.g. via finetuning) or altering how the model is used at deployment time. We focus on the latter, since information-gathering tasks rarely provide task-specific data upfront (e.g. a user's unknown preferences), and deployment-time methods avoid the cost and difficulty of finetuning an LLM altogether and are applicable to any existing LLM. However, we emphasize that improvements at the model level (e.g. Zhang et al., 2024) would be complementary to our approach.

To formalize the notion of information gathering, we need a concrete idea of what we wish to learn about. We denote the target quantity of interest as $\theta$, which may represent, for example, a user's preferences, the answer to a question, or a desired piece of content. We start with incomplete information about $\theta$, as represented by an initial *belief distribution* or prior, $p(\theta)$, but can refine these beliefs by making queries, $x \in \mathcal{X}$, to the user or some other external agent and receiving responses, $y \in \mathcal{Y}$, that are informative about $\theta$. Multiple such queries, $x_1, \ldots, x_T$, can be adaptively selected in a sequential decision-making process where we iteratively choose each $x_t$ based on the collected history, $h_{t-1} := (x_i, y_i)_{i=1}^{t-1}$. As our history grows, we will update our belief distribution to obtain $p(\theta; h_{t-1})$ via some model updating procedure.[1] In the LLM setting, there is considerable flexibility in how $p(\theta; h_{t-1})$ is constructed, as discussed in §3.1 and §4. While $p(\theta; h_{t-1})$ need not be explicitly defined, it provides the foundation for our information-theoretic method of query selection.

For clarity of exposition, we focus on the case where the $x_t$ correspond to explicit questions asked to the user, but emphasize that the approach applies more broadly to other forms of external interaction by the LLM, such as retrieving documents or calling external functions.

### 2.1 IN-CONTEXT UPDATING OF BELIEFS

A natural and cheap way to incorporate the interaction history into the LLM is to include it in the context (Brown et al., 2020). If the LLM's distribution over generated text, $z \in \mathcal{Z}$, is $p_{\mathrm{LLM}}(z)$ given appropriate prompting, then $p_{\mathrm{LLM}}(z; h_{t-1})$ is an updated distribution with the previous question–response pairs in context. From this, we can derive an updated belief distribution over $\theta$. Most simply, this can be done by using $p_{\mathrm{LLM}}(z; h_{t-1})$ to directly query about $\theta$ (e.g. if $\theta$ is some preference, we could prompt the LLM to predict this preference). However, as we show later, this approach often fails to appropriately incorporate the information from $h_{t-1}$, leading to a belief distribution inconsistent with past observations. This is consistent with recent work that shows that in context updating does not treat all contextual information equally (Kossen et al., 2024; Liu et al., 2024b; Zhang et al., 2024). In §3.1, we introduce a more robust method for deriving $p(\theta; h_{t-1})$.

---

[1] We carefully distinguish between explicit probabilistic conditioning, i.e. $p(a|b)$, and more general dependency, $p(a; b)$. The former corresponds to the conditional distribution of an associated joint distribution, $p(a, b)$, while the latter may not. Here, $h_{t-1}$ influences our distribution on $\theta$, but it is not derived via a joint distribution.

## 2.2 INFORMATION-THEORETIC EXPERIMENTAL DESIGN

The core of the BED framework is a joint generative model, $p(\theta, y; x)$, over the target quantity, $\theta$, and the outcome, $y$, corresponding to a design, $x$. Typically this is specified as a Bayesian model using a prior, $p(\theta)$, and a likelihood, $p(y|\theta; x)$. In the general case, designs are chosen to maximize the expectation of some utility function, $U(\theta, y, x)$, under this model: we choose $x^* = \arg\max_x \mathbb{E}_{p(\theta, y; x)}[U(\theta, y, x)]$. The most common choice is to take $U(\theta, y, x) = \log p(\theta, y; x) - \log p(\theta) \log p(y; x)$, where $p(\theta)$ and $p(y; x)$ are the marginal distributions on $\theta$ and $y$ implied by our joint model and we have assumed that our current beliefs on $\theta$ are independent of the design, $x$. This leads to an objective corresponding to the expected information gain (EIG) in $\theta$ (Lindley, 1956; 1972),

$$\text{EIG}_\theta(x) = \text{H}[p(\theta)] - \mathbb{E}_{p(y;x)}[\text{H}[p(\theta|y; x)]] \tag{1}$$

$$= \text{H}[p(y; x)] - \mathbb{E}_{p(\theta)}[\text{H}[p(y|\theta; x)]], \tag{2}$$

where H denotes the Shannon entropy (i.e. $\text{H}[p(\theta)] = -\mathbb{E}_{p(\theta)}[\log p(\theta)]$). We can thus equivalently think of the EIG as (a) the mutual information between $\theta$ and $y$, (b) the expected reduction in entropy over $\theta$ (i.e. information gain in $\theta$) from observing data simulated from our model, or (c) the expected reduction in entropy over data from observing $\theta$ simulated from our prior (Sebastiani & Wynn, 2000).

Working with the EIG is highly suited to a *sequential* or *adaptive* design approach, generally referred to as sequential BED or Bayesian adaptive design (Rainforth et al., 2024). Because the EIG is only a function of our underlying model, when we update the model as new data becomes available, our EIG design objective will naturally update as well. Specifically, to derive the *incremental* EIG (Cavagnaro et al., 2010) for the $t$-th query, $\text{EIG}_\theta(x_t; h_{t-1})$, we simply replace the joint in the above formulation, $p(\theta, y; x)$, with the updated joint, $p(\theta, y_t; h_{t-1}, x_t)$, with all marginals and conditionals derived from this (e.g. $p(y; x)$ becomes $p(y_t; h_{t-1}, x_t)$). Here this updated joint conventionally comes from a Bayesian update of the original model. However, in many cases, this is not practical and other non-Bayesian updates are performed instead. For example, in active learning the update often actually corresponds to retraining the model with the new data (Bickford Smith et al., 2023; Gal et al., 2017).

## 3 SEQUENTIAL BAYESIAN EXPERIMENTAL DESIGN WITH LLMS

The sequential BED framework described in §2.2 requires two core components to be specified by the user: (a) an initial joint model, $p(\theta, y; x)$, over hypotheses, $\theta$, and outcomes, $y$, given a design, $x$; and (b) a procedure to derive an updated model, $p(\theta, y_t; h_{t-1}, x_t)$, after observing $h_{t-1}$. In the LLM setting, there is significant flexibility in these critical methodological decisions. In particular, there are many ways to derive a suitable joint distribution from the LLM and its ability to learn in-context provides opportunities for update methods that go beyond standard Bayesian model updates.

**Model construction**   A major challenge in the LLM setting is that unlike conventional probabilistic models, in general, $p_{\text{LLM}}(\theta)\,p_{\text{LLM}}(y; [\theta, x]) \neq p_{\text{LLM}}(y; x)\,p_{\text{LLM}}(\theta; [x, y])$. That is, we induce a different joint distribution if we first sample $\theta$ then sample $y$ with $\theta$ in context (which we refer to as the *prior–likelihood pairing*) versus if we first sample $y$ then sample $\theta$ with $y$ in context (*data–estimation pairing*). Moreover, we can deviate from the distribution directly induced by the LLM on one or both variables. The success of using BED with LLMs turns out to be critically dependent on these choices.

We delay proper discussion of this complex issue until §4, where we will see that the preferable setup can depend on problem setting and, in particular, the relative complexity of spaces of $\theta$ and $y$. For now, we will focus on using the prior–likelihood pairing; we will argue in §4 that this is the advantageous setup in many practical scenarios. While we will generally use the LLM's directly induced distribution for the likelihood, we allow the prior to deviate from this in a problem-specific manner. As such, our initial joint model will be $p(\theta, y; x) = p(\theta)p_{\text{LLM}}(y; [\theta, x])$.

**Model updating**   Optimally updating the joint model in this setting requires incorporating new observations in a way that both fully captures the information from new data and is computationally tractable. At one extreme, we could target full Bayesian updates via approximate inference, as in classical sequential BED. However, this demands a prohibitively large number of LLM evaluations to accurately approximate the posterior, and it does not exploit the power of the LLM as a probabilistic generative model, where autoregressive sequential rollouts often lead to more nuanced and diverse behavior than repeated static queries. At the other extreme, simple in-context updating, with $p(\theta; h_{t-1}) = p_{\text{LLM}}(\theta; h_{t-1})$, is cheap but, as we show later, fails to reliably capture information

from new data, leading to inconsistent belief states and undermining the sequential BED approach. As we discuss in §3.1, we therefore employ a strategy that is somewhere between the two: drawing samples in a way that utilizes $p_{\text{LLM}}(\theta; h_{t-1})$ while encouraging diversity, then filtering out samples that are actually not compatible with $h_{t-1}$ and renormalizing. We refer to the resulting distribution as $p_f(\theta; h_{t-1})$. We do not update our likelihood model, $p_{\text{LLM}}(\theta; h_{t-1})$; see App. F.3 for empirical comparison of updated vs. static likelihoods in our experiments, and a discussion.

**BED-LLM**  We now introduce our specific algorithmic approach, BED-LLM. Here, the queries will correspond to our designs, $x$ (assumed to be in form of questions posed to the user in the following for simplicity, but could also be external function calling, document retrieval, web search, etc), and the responses received will correspond to our outcomes, $y$. Using the LLM to derive joint models over these outcomes and the target variable, $\theta$, given the history, $h_{t-1}$, as described above, we can interleave choosing informative questions by optimizing the incremental EIG, $\text{EIG}_\theta(x_t; h_{t-1})$, and updating our underlying model based on the received question–response pairs. Specifically, BED-LLM iterates over the following key steps, where $t$ indexes the current turn (see also Fig. 1).

(A) **Extract beliefs (§3.1).**  Use joint model $p(\theta, y_t; h_{t-1}, x_t) = p(\theta)p_{\text{LLM}}(y_t; [\theta, x_t])$, where $h_0 = \emptyset$, for $t = 1$ and $p(\theta, y_t; h_{t-1}, x_t) = p_f(\theta; h_{t-1})p_{\text{LLM}}(y_t; [\theta, x_t])$ for $t > 1$.

(B) **Generate candidate questions (§3.2).** Propose a candidate set of $M$ diverse, multiple-choice questions, $\mathcal{X}^{\text{cand}}$, by appropriate sampling of the LLM based on the history, $h_{t-1}$.

(C) **Estimate EIG (§3.3).** For each candidate, $x_t \in \mathcal{X}^{\text{cand}}$, estimate $\text{EIG}_\theta(x_t; h_{t-1})$.

(D) **Select and ask the best question.** Choose $x_t$ by EIG maximization and pose to the user.

(E) **Update the history.** Observe response, $y_t$, and update the history to $h_t = (h_{t-1}, (x_t, y_t))$.

## 3.1 PRIOR CONSTRUCTION AND BELIEF UPDATING

The Savage axioms (Savage, 1954) tell us that a rational agent should update its beliefs in a Bayesian manner. However, doing full Bayesian updates to our model as the history grows is generally impractical for computational reasons in the LLM setting, as it requires approximate inference and this, in turn, typically requires large numbers of expensive likelihood evaluations. Furthermore, the Savage axioms only hold if our (implied) prior truly represents our beliefs, but we find that $p_{\text{LLM}}(\theta)$ is typically heavily overconfident on a small number of possible hypotheses and can struggle to convey the full range of possibilities even with careful prompting and a high temperature (see Fig. 5).

A natural tractable alternative is to derive our beliefs through LLM in-context updates, that is, use $p_{\text{LLM}}(\theta; h_{t-1})$, noting that this has been shown to behave differently to Bayesian updating (Falck et al., 2024; Kossen et al., 2024). However, we find that even state-of-the-art LLMs such as GPT-4o (OpenAI, 2024) often fail to incorporate history faithfully; they regularly sample hypotheses incompatible with past observations and exhibit premature overconfidence, with both issues becoming more pronounced as $h_{t-1}$ grows. We discuss reasons for these shortfalls in App. A.3.

To avoid these shortfalls, we instead propose an approach that balances tractability and faithfulness. Although we will still use $p_{\text{LLM}}(\theta; h_{t-1})$ as the basis for deriving our belief state over $\theta$ (i.e. our intermediate prior), we make various alterations to effectively incorporate historical information and ensure diversity. Our derived distribution, which we refer to as $p_f(\theta; h_{t-1})$, differs from $p_{\text{LLM}}(\theta; h_{t-1})$ in two key ways. First, we filter the generated hypotheses according to whether they are compatible with the history $h_{t-1}$. We do this by using the LLM to check the compatibility of each sampled $\theta$ with all the previous question–answer pairs in $h_{t-1}$ (using $p_{\text{LLM}}(y_i; [\theta, x_i])$ for $i \in (1, 2, \ldots, t-1)$) and then rejecting that sample if an incompatibility is found. Specifically, a sample is rejected if the likelihood of an observed answer falls below a predefined threshold, chosen to balance robustness to model uncertainty against the need to enforce strict historical coherence. To reduce the computational cost of generating and evaluating hypotheses, we further include a *hypothesis-retention mechanism*: any hypotheses from the previous turn which remain consistent with the most recent question and observation are retained in the hypothesis set without regeneration.

Second, we make a number of modifications to promote diversity. Rather than generate candidates independently, we prompt the LLM to generate batches of candidates using a prompt encouraging diversity. After filtering these candidates as above and removing duplicates, we then impose a uniform distribution. Details of our exact setup for $p_f(\theta; h_{t-1})$ are given in App. E.

Current history: $h_2 = ((\text{``Born in the 20th century?''}, \text{``Yes''}), (\text{``Is this person male?''}, \text{``Yes''})).$

(A) **Extract beliefs (§3.1).** Construct hypothesis set $\Theta^{\text{cand}} = \{\theta_n\}_{n=1}^N$ by sampling candidate hypotheses, $\theta \sim p_{\text{LLM}}(\theta; h_2)$, and rejecting any $\theta$ that is inconsistent with $h_2$. Use joint model $p(\theta, y; h_2, x) = p_f(\theta; h_2)p_{\text{LLM}}(y; [\theta, x])$, where $p_f(\theta; h_2)$ is uniform over $\Theta^{\text{cand}}$.



Barack Obama     Steve Irwin     Hugh Laurie     Banksy     Elvis Presley     ...



(B) **Generate candidate questions (§3.2).** Sample $\tilde{x}_{1:M} \sim p_{\text{LLM}}(\tilde{x}_{1:M}; [h_{t-1}, \Theta^{\text{cand}}])$.

$\tilde{x}_1$: "Was this person born in Antarctica?"
$\tilde{x}_2$: "Was this person born in the 19th century?"
$\tilde{x}_3$: "Is this person an artist?"
$\tilde{x}_4$: "Is this person European?"
$\tilde{x}_5$: "Does this person prefer thrash metal over death metal?"

(C) **Estimate EIG (§3.3).** Compute $\underbrace{H[\hat{p}(y; [h_2, x])]}_{\text{marginal entropy}} - \underbrace{\frac{1}{N}\sum_{n=1}^N H[p_{\text{LLM}}(y; [\theta_n, x])]}_{\text{expected conditional entropy}} \approx \text{EIG}_\theta(x; h_2).$

|  | Question | Marg. | Exp. cond. | EIG | Intuition |
|---|---|---|---|---|---|
| $\tilde{x}_1$ | Born in Antarctica? | $\approx 0$ | $\approx 0$ | $\approx 0$ | Answer "No" for all $\theta_n$ |
| $\tilde{x}_2$ | Born in 19th century? | $\approx 0$ | $\approx 0$ | $\approx 0$ | Redundant given $h_2$ |
| $\tilde{x}_3$ | An artist? | 0.56 | 0.41 | 0.15 | Uneven split |
| $\tilde{x}_4$ | European? | 0.97 | 0.17 | 0.80 | Balanced split; crisp answers |
| $\tilde{x}_5$ | Thrash vs. death metal? | 0.89 | 0.88 | 0.01 | Uncertain even given $\theta$ |

(D) **Select and ask the best question.** Choose $x_3 = \tilde{x}_4$ by maximizing EIG. Output this to the user.

(E) **Update the history.** Observe answer, $y_3$. Set $h_3 = (h_2, (x_3, y_3))$.

Figure 1: BED-LLM applied to the 20 Questions game involves repeatedly constructing a belief state, generating candidate questions, estimating and maximizing EIG to select a question, and interacting with the user to gather new data. The contrast between $\tilde{x}_4$ and $\tilde{x}_5$ illustrates the benefit of using EIG with a non-deterministic likelihood: $\tilde{x}_5$ has high marginal entropy (answers are uncertain), but its expected conditional entropy is equally high (answers are uncertain even given $\theta$), so nearly nothing is expected to be learned, and $\tilde{x}_4$ should be favoured as a result. Numerical values are illustrative.

## 3.2 GENERATING CANDIDATE QUESTIONS

As it is not computationally feasible to directly optimize over the space of possible questions, we rely on using the LLM to propose diverse candidate questions, $\mathcal{X}^{\text{cand}}$, then select the best question from these. We consider two specific approaches. (a) *Unconstrained generation:* given $h_{t-1}$, the LLM is simply asked to propose new questions by sampling from $p_{\text{LLM}}(x_t; h_{t-1})$ with appropriate prompting. (b) *Conditional generation:* the LLM is given both $h_{t-1}$ *and* a generated set of hypotheses, $\Theta^{\text{cand}} = \{\theta_n\}_{n=1}^N$, such that we sample from $p_{\text{LLM}}(x_t; [h_{t-1}, \Theta^{\text{cand}}])$; specifically, the LLM is prompted for questions that "slice" the hypothesis pool into roughly balanced subsets.

For both strategies, we sample $M$ questions jointly with a relatively high temperature to encourage diversity. Conditional generation allows us to "guide" the LLM to propose highly informative questions. However, it risks overfitting to $\Theta^{\text{cand}}$. In practice, we find it is effective for discrete spaces (§6.1), but less so for spaces with complex, overlapping hypotheses (§6.2). We restrict questions to multiple-choice format to simplify uncertainty quantification (see §4).

## 3.3 ESTIMATING EIG FOR EACH QUESTION

To estimate the EIG based on Equation 2 for a given question $x_t$, we use the following Rao-Blackwellized estimator based on the LLM's predictive distribution:

$$\text{EIG}_\theta(x_t; h_{t-1}) \approx \frac{1}{N}\sum_{n=1}^N \sum_{y_t \in \mathcal{Y}} p_{\text{LLM}}(y_t; [\theta_n, x_t]) \log p_{\text{LLM}}(y_t; [\theta_n, x_t])$$
$$- \sum_{y_t \in \mathcal{Y}} \hat{p}(y_t; [h_{t-1}, x_t]) \log \hat{p}(y_t; [h_{t-1}, x_t]), \tag{3}$$

where $\hat{p}(y_t; [h_{t-1}, x_t]) := \frac{1}{N} \sum_{n=1}^{N} p_{\text{LLM}}(y_t; [\theta_n, x_t])$ and $\theta_n \sim p_f(\theta; h_{t-1})$ (see §3.1). This estimator has been used in other BED contexts (Gal et al., 2017; Rainforth, 2017). Note that the samples do not need to be independent for this estimator to converge, provided they satisfy some appropriate form of ergodicity or decaying correlation (see, e.g., Billingsley (2013)). When constructing this estimator, we compute the $p_{\text{LLM}}(y_t; [\theta_n, x_t])$ terms using the LLM's logits whenever possible. By the Rao-Blackwell theorem, this always produces lower variance than purely sample-based estimators (Rao et al., 1945), like those employed in Hu et al. (2024) and Kobalczyk et al. (2025).

**Avoiding deterministic likelihood assumptions**   Previous attempts to apply information criteria to choosing queries in LLMs have generally assumed responses are deterministic given $(\theta, x_t)$ (Cooper et al., 2025; Kobalczyk et al., 2025; Hu et al., 2024; Mazzaccara et al., 2024; Piriyakulkij et al., 2023). Under this assumption, the EIG simplifies to the marginal predictive entropy, $\text{H}[p(y_t; x_t, h_{t-1})]$.

This is problematic as, in practice, the expected likelihood entropy will vary with $x_t$. In general, $\mathbb{E}_{p(\theta; h_{t-1})}[\text{H}[p(y_t|\theta; x_t, h_{t-1})]]$ measures how certainly the question can be answered once $\theta$ is known. Including it in our objective is essential in avoiding questions that are irrelevant, ambiguous, unclear, or simply unhelpful in our quest to learn about $\theta$. Fig. 1 illustrates this concretely—question $\tilde{x}_4$ has high marginal entropy and low conditional entropy (answers are crisp given $\theta$), yielding high EIG, while $\tilde{x}_5$ has equally high marginal entropy but also high conditional entropy (answers are noisy *even given* $\theta$), so the terms cancel and nearly nothing is learned. An entropy-only method would score both similarly and risk wasting a turn; the same failure mode causes the Entropy ablation to provide no improvement over Prompt-Only in our preference elicitation experiments (§6.2). Given that both terms of Eq. 3 are computed from the same LLM likelihood evaluations, retaining the full EIG requires no additional calls and provides no computational savings. Thus, we advise against making deterministic likelihood assumptions.

## 4   ON THE SPECIFICATION OF $p(\theta, y_t; h_{t-1}, x_t)$, AND ITS IMPLICATIONS

As we described in §3, successfully applying sequential BED in the LLM setting hinges upon how we specify, and update, the joint distribution, $p(\theta, y_t; h_{t-1}, x_t)$. In particular, as previously highlighted, there are two distinct ways to derive the joint model from our LLM: using a *prior–likelihood pairing*, $p(\theta; h_{t-1}) p(y_t; [\theta, x_t])$, or a *data–estimation pairing*, $p(y_t; [h_{t-1}, x_t]) p(\theta; [h_{t-1}, x_t, y_t])$. The first construction mirrors deriving our beliefs about $\theta$ from a *conventional* Bayesian posterior with a concrete prior and likelihood derived (at least partially) from the LLM, whereas the second has analogies to a *marginal-posterior* approach (Fong et al., 2023; Falck et al., 2024) in that it that samples hypothetical data and draws inferences on $\theta$ given hypothetical data using in-context learning. In our outlined BED-LLM approach, we adopted a prior–likelihood pairing. Below, we justify this decision and also discuss certain settings where the data–estimation setup might be preferable instead.

**Modeling flexibility**   The most obvious relative merits of the prior–likelihood and data–estimation pairings are in the flexibility in how each term is chosen. The prior–likelihood pairing gives us greater flexibility to construct a prior set of beliefs over $\theta$ that is distinct to the LLM's internal beliefs, as it allows us to directly control this prior by changing $p(\theta; h_{t-1})$, whereas the prior is only implicitly defined in the data–estimation pairing. In §3.1 we exploited this flexibility through our definition of $p_f(\theta; h_{t-1})$. On the other hand, the data–estimation pairing could provide some beneficial flexibility in specifying how the data itself is simulated through changing $p(y_t; [h_{t-1}, x_t])$, which could, for example, be useful when we have access to external data simulators.

**Faithfulness of conditional distributions**   While deviations from relying on direct LLM predictions are also in principle possible for the conditional models $p(y_t; [\theta, x_t])$ and $p(\theta; [h_{t-1}, x_t, y_t])$, in practice, these will typically be more difficult and expensive to implement. This is first because these conditionals need to be instantiated for each sampled instance of the conditional variable ($\theta$ and $y_t$ respectively), rather than just needing us to set up a single marginal distribution. Second, to construct estimators for Equations (1) and (2), we require access to concrete *probabilities* for the conditional distributions (in order to calculate entropies), whereas we only needed to draw samples for the marginal distributions (in order to approximate expectations). As such, the conditionals need to be explicit distributions, or at least ones where the probability can be cheaply estimated, so they are more difficult to define through the output of some algorithmic procedure, especially in large spaces.

When considering the conditional distributions, the decisive question on the relative merit of the two formulations is which conditional factor we are willing to trust the LLM to supply as a *full probability distribution*. Critically, we rely on how the LLM captures uncertainty in this full distribution— including, for example, tail behavior—not merely the fidelity of typical samples; the marginal factors, by contrast, only need to be sampled from. If we accept the LLM's direct predictive distribution for $p(y_t; [\theta, x_t])$, then we are basing our notion of uncertainty around (and will need to calculate) $\mathrm{H}[p_{\mathrm{LLM}}(y_t; [\theta, x_t])]$, and if instead we place more faith in the LLM's internal distribution for $p(\theta; [h_{t-1}, x_t, y_t])$, then we are basing our uncertainty around $\mathrm{H}[p_{\mathrm{LLM}}(\theta; [h_{t-1}, x_{t+1}, y_{t+1}])]$. In essence, the choice between prior–likelihood and data–estimation pairings thus comes down to whether we believe the LLM will produce a more appropriate conditional uncertainty over $\theta$ or $y$, along with our ability to numerically estimate this uncertainty cheaply.

This difference becomes particularly noticeable when the complexities of the spaces of $\theta$ and $y$ differ significantly. Our ability to draw sensible samples of either will generally be quite robust to these spaces being complex or high-dimensional; this is where LLMs tend to thrive, effectively generating highly complex outputs in an autoregressive manner. However, evaluating the entropy of a distribution becomes dramatically harder as the dimensionality or complexity increases (Acharya et al., 2019; Paninski, 2003), and the entropy of the predictive distribution of an LLM in such cases will *not* typically provide a sensible measure of uncertainty (Kadavath et al., 2022; Desai & Durrett, 2020). As such, the decision between joint formulations should predominantly be based on the complexity of the space of $\theta$ versus that of $y$: *we should generally favor the prior–likelihood formulation if $\theta$ is more complex and the data–estimation formulation if $y$ is more complex.* For the problems that we consider, the space of $y$ is less complex than that of $\theta$, indicating we should, in general, use the prior–likelihood formulation. However, in cases where this is not true, the data–estimation formulation may be preferable instead, see App. B for further discussion on this and on the choice of $\theta$.

**Extracting the belief state** A further advantage of the prior–likelihood construction is that our belief state on $\theta$ can be extracted directly as $p_f(\theta; h_{t-1})$. With the data–estimation construction, we would have to estimate the marginal on $\theta$ by integrating $p(y_t; [h_{t-1}, x_t]) \, p(\theta; [h_{t-1}, x_t, y_t])$ over the synthetic response $y_t$. Direct access to $p(\theta; h_{t-1})$ is also important to ensure that our current belief state is independent of the next question $x_t$, which is both intuitively desirable and theoretically required to be a valid BED approach (Lindley, 1972); data–estimation formulations will generally violate this.

## 5 RELATED WORK

Several works have explored the baseline ability of LLMs to rapidly learn about a parameter of interest by asking questions (Zhang et al., 2024; Li et al., 2025b)—effectively our *Prompt-Only* baseline in §6. While these works demonstrate some ability to adaptively construct information-seeking questions, they often fail to extract important information (Li et al., 2025a).

Some works have further specifically attempted to choose questions based on model-based uncertainty criteria (Piriyakulkij et al., 2023; Hu et al., 2024; Kobalczyk et al., 2025; Mazzaccara et al., 2024; Cooper et al., 2025). None of these works provide the same careful consideration of how the underlying joint model should be formulated, which underpins our own work, and they all assume deterministic likelihood models that mean their objectives correspond to a sample-based estimate of marginal predictive entropy in practice, as explained in §3.3. In general, these previous works have also required restrictions on the space of allowable hypotheses, $\theta$, and typically require additional assumptions and/or approximations. More extensive discussion of related work is given in App. C.

## 6 EXPERIMENTS

We now assess how well BED-LLM and alternative information-gathering approaches work in two practical scenarios: 20 Questions, a game in which the player has to guess a target entity and can ask up to 20 yes-no questions about the entity; and preference elicitation, a task in which the agent has to predict a user's preference profile and can ask five multiple-choice questions to the user.

**Answerer** We produce answers to the *questioner* LLM's questions using a separate *answerer* LLM. The answerer is provided with a ground-truth $\theta^*$ (a target entity in 20 Questions or a user profile in preference elicitation) and processes individual questions from the questioner without access to any of the questioner's context (i.e. $h_{t-1}$ and the questioner's prompts). We test two questioner–answerer setups, where the two are served by separate instances of the same LLM, or two different LLMs. The latter sce-

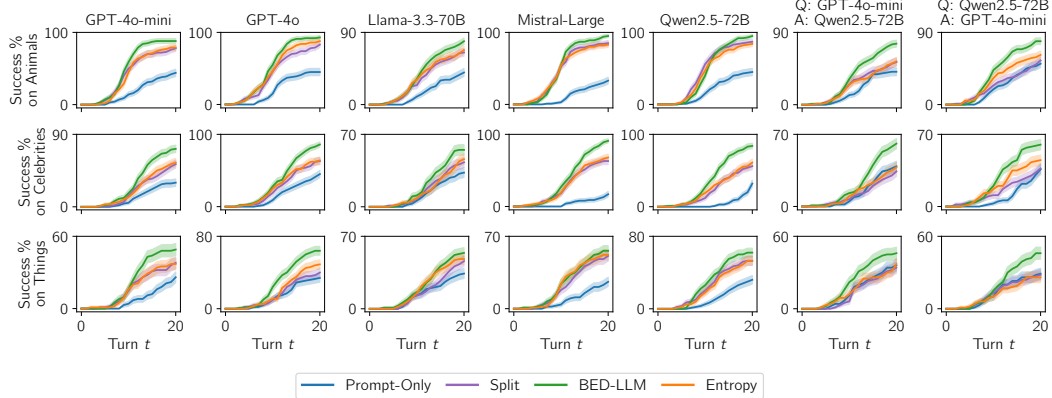

Figure 2: Success rate on 20 Questions: mean $\pm$ standard error across 100 targets per dataset.

nario is important because in practice, the answerer will often follow a different distribution than the questioner's internal model for reasoning about responses, thereby forming a model misspecification.

**Baselines** We compare BED-LLM against three baselines. *Prompt-Only* prompts the questioner to directly generate an informative next question, without explicit hypothesis generation or a data-acquisition objective, and then sampling the question with temperature $T = 1$; this was explored by Zhang et al. (2024). *CoT* augments Prompt-Only with a ReAct-style (Yao et al., 2023b) chain-of-thought: the LLM first produces a thought (what has been established, what information would help most) then an action (the question). This tests whether structured reasoning alone can close the gap to BED-LLM. *Split* chooses the question that most equally splits a sampled set of hypotheses, $\Theta^{\text{cand}}$, which corresponds to maximizing the marginal predictive entropy, $\mathrm{H}[p(y_t; x_t, h_{t-1})]$, in a model with a deterministic likelihood. As such, the methods of Cooper et al. (2025), Hu et al. (2024), Kobalczyk et al. (2025), Mazzaccara et al. (2024) and Piriyakulkij et al. (2023) can all be viewed as variants of this objective. To the best of our knowledge, Split represented the previous state-of-the-art method for 20 Questions. We note that our own specific Split baseline implementation, which uses BED-LLM's filtering mechanism, also achieves dramatically better results than reported by, for example, Kobalczyk et al. (2025), so this constitutes a strong baseline relative to previous work. On top of this, our implementation of Prompt-Only appears to significantly improve over that of Zhang et al. (2024). Additional evaluations and full plots are in App. F.

## 6.1  20 QUESTIONS

We consider three sets of 20 Questions problems: Animals, Celebrities, and Things (See App. H.1). Each problem set comprises 100 target entities, $\{\theta_i^*\}_{i=1}^{100}$. The space of possible $\theta$ is large and not explicitly defined or restricted: we do not tell the LLM this set of target entities, so the space is bounded only by what the LLM can generate; by comparison, many previous works have relied on restricted spaces for $\theta$ (Chan et al., 2025; Hu et al., 2024; Piriyakulkij et al., 2023; Wang et al., 2025).

To evaluate performance, at each turn, $t \in (0, 1, \ldots, 20)$, we extract $\theta_i^t$ from $p_f(\theta_i; h_t)$ using greedy decoding and we compute the success rate as the mean across $i$ of $\mathbb{I}(\theta_i^t = \theta_i^*)$. These evaluation guesses are not part of the questioner algorithm itself and are not included in $h_{t-1}$. In line with the original rules of the game, we also introduce an explicit mechanism for the questioner to guess the answer as one of its 20 questions: if the set of filtered hypotheses collapses to a single candidate, the questioner asks "Is it ⟨item⟩?". A correct guess ends the game; otherwise the negative response is added to $h_{t-1}$ and counted towards the budget. See App. H for further experimental details.

**BED-LLM improves over Prompt-Only and Split baselines** Our results in Fig. 2 and Tab. 1 show BED-LLM significantly outperforming both baselines across all problems and LLMs. Particularly notable is that BED-LLM's final success rate is typically more than double that of Prompt-Only, highlighting the substantial gains that can be achieved by using explicit EIG maximization.

**BED-LLM ablations** We further evaluate four ablations of BED-LLM to isolate the contribution of each of its core components; full descriptions are in App. G. *Entropy* replaces the full EIG objective with the marginal predictive entropy, $\mathrm{H}[p(y_t; x_t, h_{t-1})]$; it is similar to the **Split** baseline, but uses BED-LLM's likelihood model instead of a deterministic one. *Data–Estimation* swaps BED-LLM's

| | | Success Rate (%) | | | | | | | |
|---|---|---|---|---|---|---|---|---|---|
| | **Model** | **Prompt-Only** | **Split** | **CoT** | **BED-LLM** | **Entropy** | **Data–Est.** | **ICL Beliefs** | **Impl. Max.** |
| Animals | GPT-4o-mini | $44_{\pm5.0}$ | $78_{\pm4.2}$ | $55_{\pm5.0}$ | $\mathbf{88}_{\pm3.3}$ | $79_{\pm4.1}$ | $64_{\pm4.8}$ | $18_{\pm3.9}$ | $47_{\pm5.0}$ |
| | GPT-4o | $45_{\pm5.0}$ | $83_{\pm3.8}$ | $62_{\pm4.9}$ | $\mathbf{93}_{\pm2.6}$ | $88_{\pm3.3}$ | $70_{\pm4.6}$ | $25_{\pm4.4}$ | $70_{\pm4.6}$ |
| | Llama-3.1-8B | $8_{\pm2.7}$ | $49_{\pm5.0}$ | $19_{\pm3.9}$ | $\mathbf{63}_{\pm4.9}$ | $54_{\pm5.0}$ | $38_{\pm4.9}$ | $25_{\pm4.4}$ | $16_{\pm3.7}$ |
| | Llama-3.3-70B | $40_{\pm4.9}$ | $65_{\pm4.8}$ | $40_{\pm4.9}$ | $\mathbf{79}_{\pm4.1}$ | $68_{\pm4.7}$ | $40_{\pm4.9}$ | $33_{\pm4.7}$ | $54_{\pm5.0}$ |
| | Mistral-Large | $33_{\pm4.7}$ | $85_{\pm3.6}$ | $35_{\pm4.8}$ | $\mathbf{95}_{\pm2.2}$ | $83_{\pm3.8}$ | $83_{\pm3.8}$ | $53_{\pm5.0}$ | $53_{\pm5.0}$ |
| | Qwen2.5-72B | $45_{\pm5.0}$ | $87_{\pm3.4}$ | $51_{\pm5.0}$ | $\mathbf{95}_{\pm2.2}$ | $85_{\pm3.6}$ | $68_{\pm4.7}$ | $46_{\pm5.0}$ | $61_{\pm4.9}$ |
| Celebrities | GPT-4o-mini | $30_{\pm4.6}$ | $53_{\pm5.0}$ | $42_{\pm5.0}$ | $\mathbf{72}_{\pm4.5}$ | $55_{\pm5.0}$ | $32_{\pm4.7}$ | $16_{\pm3.7}$ | $31_{\pm4.7}$ |
| | GPT-4o | $45_{\pm5.0}$ | $63_{\pm4.9}$ | $63_{\pm4.9}$ | $\mathbf{86}_{\pm3.5}$ | $64_{\pm4.8}$ | $55_{\pm5.0}$ | $52_{\pm5.0}$ | $50_{\pm5.0}$ |
| | Llama-3.1-8B | $10_{\pm3.0}$ | $35_{\pm4.8}$ | $16_{\pm3.7}$ | $\mathbf{58}_{\pm5.0}$ | $36_{\pm4.8}$ | $19_{\pm3.9}$ | $24_{\pm4.3}$ | $19_{\pm3.9}$ |
| | Llama-3.3-70B | $33_{\pm4.7}$ | $43_{\pm5.0}$ | $36_{\pm4.8}$ | $\mathbf{55}_{\pm5.0}$ | $46_{\pm5.0}$ | $26_{\pm4.4}$ | $27_{\pm4.5}$ | $37_{\pm4.9}$ |
| | Mistral-Large | $19_{\pm4.0}$ | $63_{\pm4.9}$ | $42_{\pm5.0}$ | $\mathbf{91}_{\pm2.9}$ | $68_{\pm4.7}$ | $66_{\pm4.8}$ | $31_{\pm4.7}$ | $36_{\pm4.8}$ |
| | Qwen2.5-72B | $32_{\pm4.7}$ | $56_{\pm5.0}$ | $48_{\pm5.0}$ | $\mathbf{84}_{\pm3.7}$ | $59_{\pm4.9}$ | $34_{\pm4.8}$ | $26_{\pm4.4}$ | $39_{\pm4.9}$ |
| Things | GPT-4o-mini | $26_{\pm4.4}$ | $38_{\pm4.9}$ | $33_{\pm4.7}$ | $\mathbf{49}_{\pm5.0}$ | $37_{\pm4.9}$ | $26_{\pm4.4}$ | $19_{\pm4.0}$ | $25_{\pm4.4}$ |
| | GPT-4o | $34_{\pm4.8}$ | $40_{\pm4.9}$ | $49_{\pm5.0}$ | $\mathbf{64}_{\pm4.8}$ | $49_{\pm5.0}$ | $26_{\pm4.4}$ | $19_{\pm3.9}$ | $42_{\pm5.0}$ |
| | Llama-3.1-8B | $10_{\pm3.0}$ | $12_{\pm3.3}$ | $10_{\pm3.0}$ | $\mathbf{26}_{\pm4.4}$ | $15_{\pm3.6}$ | $9_{\pm2.9}$ | $11_{\pm3.1}$ | $10_{\pm3.0}$ |
| | Llama-3.3-70B | $34_{\pm4.8}$ | $46_{\pm5.0}$ | $35_{\pm4.8}$ | $\mathbf{55}_{\pm5.0}$ | $48_{\pm5.0}$ | $19_{\pm3.9}$ | $15_{\pm3.6}$ | $34_{\pm4.8}$ |
| | Mistral-Large | $26_{\pm4.4}$ | $51_{\pm5.0}$ | $29_{\pm4.6}$ | $\mathbf{58}_{\pm5.0}$ | $52_{\pm5.0}$ | $46_{\pm5.0}$ | $19_{\pm3.9}$ | $30_{\pm4.6}$ |
| | Qwen2.5-72B | $32_{\pm4.7}$ | $51_{\pm5.0}$ | $46_{\pm5.0}$ | $\mathbf{62}_{\pm4.9}$ | $51_{\pm5.0}$ | $39_{\pm4.9}$ | $24_{\pm4.3}$ | $40_{\pm4.9}$ |

Table 1: Success rate (%) for 20 Questions at the end of the game. Best result in bold. $\pm$ numbers show the standard error of the mean estimated using $\sqrt{p(1-p)/(n-1)}$ where $p$ is the success percentage and $n$ is the number of datapoints. This estimator is positively biased and thus conservative.

| **Method** | **Joint model** | **Objective** | **Belief updates** |
|---|---|---|---|
| **BED-LLM** | Prior–likelihood | Full EIG (Eq. 3) | Filtered ($p_f$) |
| Prompt-Only | None (implicit LLM) | None (implicit LLM) | Raw ICL ($p_{\text{LLM}}$) |
| Split | Deterministic likelihood | Pred. entropy | Filtered ($p_f$) |
| Entropy | Prior–likelihood | Pred. entropy | Filtered ($p_f$) |
| Data–Estimation | Data–estimation | Full EIG | Filtered ($p_f$) |
| ICL Beliefs | Prior–likelihood | Full EIG | Raw ICL ($p_{\text{LLM}}$) |
| Implicit Max. | Prior–likelihood | LLM judgment | Filtered ($p_f$) |

Table 2: Summary of how each method relates to BED-LLM's three core algorithmic components. Each ablation (bottom section) modifies exactly one component of BED-LLM, highlighted in  grey .

prior–likelihood joint model for a data–estimation pairing (see App. D), testing the importance of BED-LLM measuring uncertainty in $y$ space instead of $\theta$ space (see §3). *ICL Beliefs* omits our filtering procedure in the belief extraction (see §3.1), just using the, often incoherent, raw in-context beliefs $p_{\text{LLM}}(\theta; h_{t-1})$ instead of $p_f(\theta; h_{t-1})$. Finally, *Implicit Maximization* replaces explicit EIG estimation with LLM judgment, drawing on the Tree of Thoughts (ToT; Yao et al., 2023a) intuition of reasoning over hypothetical futures: the model is presented with the same candidate questions as BED-LLM, but is simply prompted to internally reason and select the question it judges most informative, providing a lightweight alternative to explicit EIG estimation.

In Tab. 1 we see that BED-LLM comfortably outperforms all alternative approaches. Notably, Entropy provides the strongest alternative, with performance slightly better than Split. The fact that Entropy performs much more similarly to Split than BED-LLM shows that the use of a non-deterministic likelihood is beneficial predominantly in allowing us to target a proper EIG, rather than due to changes in the marginal predictive entropy itself. The improvement of Implicit Maximization over Prompt-only is also notable, given how cheap it is to run (see Tab. 3 for run time information).

**Prior–likelihood outperforms data–estimation** Our analysis in §4 is validated by our results: BED-LLM's prior-likelihood approach substantially outperforms Data–Estimation. Data–Estimation still outperforms Prompt-Only, but interestingly it performs worse than Entropy, highlighting the importance of estimating uncertainty in the $y$ space instead of $\theta$ space here. These findings reinforce our claim that the choice of joint-model factorization is a critical algorithmic decision.

Figure 3: Mean rating across 10 film recommendations: mean $\pm$ standard error across 200 users.

**Rejection sampling and explicit EIG maximization are key** We also see how we produce our beliefs over $\theta$ matters: deriving beliefs using simple in-context learning, as in ICL Beliefs, lead to massive performance drops. Further, while BED-LLM's routines for sampling candidate questions and hypotheses are crucial, they alone are not sufficient: passing the samples to an LLM and prompting it to select the highest-EIG question, as in Implicit Maximization, works much less well.

**BED-LLM is robust to questioner–answerer mismatch** Our results in Fig. 2 demonstrate that the benefit of BED-LLM persists even under model misspecification. This is important for applicability to real-world users, whose responses will follow a different distribution than the questioner LLM.

## 6.2 Preference Elicitation

Unlike 20 Questions, in which $\theta$ is a concrete entity and most reasonable questions have clear answers, many real-world information-gathering tasks involve more abstract targets and less predictable data generation. A key example is learning user preferences, where it may be difficult to explicitly define a concrete closed set of possible $\theta$, or for the LLM to develop appropriate uncertainty estimates. To study such a scenario, we evaluate BED-LLM on inferring users' film preferences. Here the target $\theta$ is somewhat abstract, and we have more flexibility in how we define it in our joint model. Our chosen setup is to define $\theta$ to be a user profile, namely a paragraph of text describing the user's film tastes with our answerer model prompted to emulate a user with a given profile; see App. I for full details. We consider 200 ground-truth profiles, $\{\theta_i^*\}_{i=1}^{200}$, which, as with 20 Questions, are never revealed to the questioner. Because Split is not applicable as a baseline here (a deterministic likelihood assumption is clearly unreasonable), we benchmark with the similar Entropy approach instead. We also note that data–estimation setup is completely unviable here due to the large $\theta$ space.

At each turn, $t \in (0, 1, \dots, 5)$, the questioner uses $h_{t-1}$ to generate ten film recommendations. This list is then rated in its fit to the user profile using an LLM-as-judge setup (Trivedi et al., 2024; Zhu et al., 2025). Specifically, the answerer scores each film on a scale of 1 to 5 (in 0.5 increments), based on how well the film aligns with $\theta^*$; this score is output together with a brief justification to increase reliability. The films' scores are not included in $h_{t-1}$.

Our results in Fig. 3 show that, while Prompt-Only seems to be a stronger baseline in this preference-elicitation scenario than for 20 Questions, BED-LLM is still able to provide a boost over both it and Entropy, producing higher-rated film recommendations. BED-LLM's benefit is most clear in scenarios where the questioner belongs to a different model class to the answerer.

## 7 Conclusion

In this work, we have shown how to effectively apply the framework of sequential Bayesian experimental design (BED) to the problem of interactive information gathering with LLMs. Specifically, we have introduced BED-LLM, which provides a specific, information-theoretic, sequential BED approach that makes a variety of carefully justified design choices in the joint-model factorization, belief updating, and EIG estimation. Particularly central to BED-LLM is the prior–likelihood pairing with filtering of hypotheses for consistency with the history. BED-LLM is notably the first work that uses both this prior–likelihood pairing without making a deterministic likelihood assumption that causes the EIG to simply to just marginal predictive entropy. Together, these innovations lead to substantial performance improvements compared to previous approaches. The results thus confirm that principled EIG-driven strategies can yield substantial gains for interactive, multi-turn information gathering problems.

ACKNOWLEDGMENTS

DC is supported by the EPSRC CDT in Statistics and Machine Learning (EP/Y034813/1), an Oxford-Radcliffe Scholarship and an Oxford AI Fast Exploration grant. FBS is supported by the EPSRC Probabilistic AI Hub (EP/Y028783/1). TR is supported by the UK EPSRC grant EP/Y037200/1.

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

APPENDIX CONTENTS

## A DISCUSSION RELATING TO §3

### A.1 UPDATING THE LIKELIHOOD

The success of BED-LLM hinges on our ability to update our joint distribution. As mentioned in §3, we choose not to update the likelihood model as more data is gathered, that is, our likelihood in the sequential setting will be $p_{\text{LLM}}(y_t; [\theta, x_t])$ instead of $p_{\text{LLM}}(y_t; [h_{t-1}, \theta, x_t])$. The main rationale of this choice is that for many problems our beliefs on $\theta$ capture all the required information to predict $y|x$, hence including the history is adding unnecessary context that could influence the LLM's behavior in undesirable ways. See App. F.3 for a results comparison of static and updated likelihoods on the 20 Questions game. However, it is important to note that $p_{\text{LLM}}(y_t; [h_{t-1}, \theta, x_t])$ should be used instead for problems where $\theta$ will not capture all information from previous data, e.g. if $\theta$ is a binary value corresponding to whether we reject a null hypothesis, or is the answer to a particular other question of interest.

### A.2 ESTIMATING EIG FOR EACH QUESTION

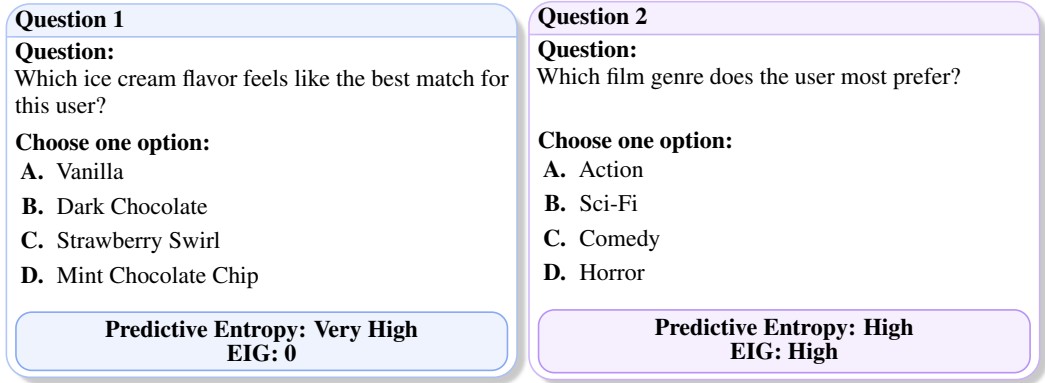

Figure 4: Predictive entropy vs. expected information gain (EIG) in a film-preferences elicitation task. Left: very high predictive entropy (answer is completely unknown) but EIG = 0 because the answer provides no insight into the user's film preferences. Right: both predictive entropy and EIG are high as the answer is uncertain, but different answers would lead to markedly different posterior updates, making it informative for learning film preferences. This thus demonstrates how the two criteria can select different questions.

#### A.2.1 PREDICTIVE ENTROPY IS NOT A GOOD APPROXIMATION OF EIG

As discussed in §3.3, previous information-based query selection mechanisms have assumed that responses are deterministic given $\theta$ and $x$. This implies that the expected entropy of the likelihood, $\mathbb{E}_{p(\theta; h_{t-1})}[\text{H}[p(y_{t+1}|\theta; x_{t+1}, h_{t-1})]]$, is constant over designs, meaning that maximizing EIG is equivalent to maximizing the marginal predictive entropy, $\text{H}[\mathbb{E}_{p(\theta; h_{t-1})}[p(y_t|\theta; x_t, h_{t-1})]]$.

In practice, the expected likelihood entropy can and will vary across designs. This variability in the expected likelihood entropy can be crucial in selecting good designs. A concrete example helps highlight how predictive entropy can differ significantly from EIG. Fig. 4 shows two candidate questions that could be asked to elicit film preference. Question 1 has high predictive entropy: in a randomly selected group of people, we would expect high variation in ice cream preference (regardless of the individual's film preferences). However, since ice cream preference is unrelated to film preference, the answer would not help us narrow down our hypothesis space, and the EIG is zero.

This is also supported by evidence in our experiments (§6). Both the Split baseline, and the Entropy ablation, assume a deterministic likelihood; in particular, the Entropy ablation uses the same estimator of the predictive entropy as BED-LLM. In both cases, we see the performance significantly degrades relative to using the full EIG. Further, omitting the expected likelihood entropy term provides no meaningful computational saving—the same LLM evaluations are used for the top and bottom lines of Eq. 3, hence doing the full estimate of the EIG requires no additional LLM calls to be made.

### A.2.2 EIG ESTIMATOR

One might be tempted to replace $\hat{p}(y_{t+1}; [h_{t-1}, x_{t+1}])$ with $p_{\mathrm{LLM}}(y_{t+1}; [h_{t-1}, x_{t+1}])$ in the EIG estimation in Eq. 3, as the two essentially offer alternative predictive distributions for the outcome. We also advise against this, though, noting that it again provides no meaningful computational benefits (unless one also assumes a deterministic likelihood, but this would then mean we no longer consider $\theta$ at all). A key reason for avoiding this substitution is that it would mean we are no longer estimating a true EIG: the inconsistency between the likelihood and the marginal data distribution means there is no longer a joint model where we are minimising our expected uncertainty in $\theta$. We also find that the LLM process of sampling $\theta$ from $p_f(\theta; h_{t-1})$ followed by $y$ from $p_{\mathrm{LLM}}(y_t; [\theta, x_t])$ tends to give a better uncertainty over responses than sampling $y$ directly from $p_{\mathrm{LLM}}(y_{t+1}; [h_{t-1}, x_{t+1}])$.

### A.3 PRIOR CONSTRUCTION AND BELIEF UPDATING

In §3.1, we argued that Prompt-Only in-context updating is not sufficient for updating our beliefs: We fail to fully incorporate the information from the history $h_t$, and we often have overconfident distributions. The shortfalls of in-context learning in such settings have also previously be noted by, for example, (Liu et al., 2024b; Zhang et al., 2025; 2024). We posit two reasons why they likely struggle in such settings. First, the information from the different examples in the history are generally highly distinct in these information-gathering settings (indeed, this is part of our aim in adaptively design informative questions), making it harder for the LLM to appropriately reconcile all the provided information than in many other uses of in–context learning. Second, $\theta$ will often represent a user–specific variable that cannot easily be predicted from any data other than the user's responses to questions: it has been argued that much of the success of in–context learning in LLMs is down to improving problem specification and linking the requested task to data it has seen in its training, rather than truly "learning" from the provided examples (Min et al., 2022; Kossen et al., 2024), but the history in our setting is rarely helpful for this due to its user–specific nature.

## B DISCUSSION RELATING TO §4

### B.1 AN ALTERNATIVE VIEW ON THE FAITHFULNESS OF CONDITIONAL DISTRIBUTIONS

Another way of viewing the distinction between the prior–likelihood and data–estimation constructions is in which of the EIG forms, Eq. 1 or Eq. 2, we center our reasoning. For a given joint model, the two are, of course, mathematically equivalent. However, they give us different ways of thinking about what it means to maximize the EIG: reducing entropy in $\theta$ from seeing $y$, or reducing entropy in $y$ from seeing $\theta$. This, in turn, gives us a way to reason about how appropriate our joint model is. When we choose to use one of $p(y_t; [\theta, x_t])$ or $p(\theta; [h_{t-1}, x_t, y_t])$, we are centering our reasoning around the entropy of this quantity making sense, while allowing the other entropy in the other form to be implicitly defined from the resulting joint distribution; because the two forms are equivalent, we know that if our explicit form is suitable/unsuitable, the implicit form will be as well. If, for example, we directly fix the form of $p(\theta; [h_{t-1}, x_t, y_t])$ using our LLM's predictive distribution, we are also directly relying on its expected entropy being a meaningful measure of design quality. If $\theta$ is high-dimensional and predominantly free-form, the resulting entropy produced by the LLM is unlikely to be meaningful and using the data–estimation pairing is unlikely to produce an effective strategy. However, if $y$ is instead quite constrained, the LLM can produce a meaningful entropy over it, and choosing a model based on the prior–likelihood pairing is likely to *implicitly define* a meaningful distribution, and thus entropy, on $\theta$. Conversely, if $\theta$ is constrained and $y$ is free form, the opposite will hold instead.

### B.2 CHOICE OF $\theta$

An important corollary of this reasoning is that it can be important to be careful in our choice of exactly what we take $\theta$ to be, especially if we are using the data–estimation formulation. In particular, it is essential for entropy in the space of $\theta$ to form a meaningful notion of uncertainty, even if this entropy is not being measured through the LLM's predictive distribution of $\theta$ directly. Thus, while $\theta$ inherently represents what we are trying to learn about and should always be set up as such, if there is flexibility in how exactly we formulate it, we should be careful to choose a form that yields

an appropriate uncertainty measure. For example, if the LLM is trying to clarify what code a user wishes it to generate, we could either choose $\theta$ to be the code itself or, following (Neiswanger et al., 2021; Bickford Smith et al., 2023), the output the code produces. Here the entropy over code outputs induced by our distribution on code is likely to be a much better measure of uncertainty than the entropy of the raw code itself, given that there are multiple ways one can code the same operation.

### B.3 Alignment between EIG and belief updating procedure

If our ultimate goal is to minimize uncertainty in $\theta$, as measured by its entropy, then we can use the expected uncertainty reduction framework of Bickford Smith et al. (2025) to provide insights into how well our EIG formulation and belief updating procedures align. To simplify discussions, for now we consider the setting where we choose a single question $x$ and obtain a response $y$. Following Bickford Smith et al. (2025), we can think of the "true" optimal design as selecting

$$x^*_{\text{true}} = \arg\min_x \mathbb{E}_{p_{\text{true}}(y;x)}\left[\text{H}[p(\theta;x,y)]\right], \tag{4}$$

where $p_{\text{true}}(y;x)$ is the true response distribution and $p(\theta;x,y)$ is our belief state after the experiment.

Note here that true optimal design has no direct dependency on our current beliefs about $\theta$; it only depends on $p_{\text{true}}(y;x)$ and the hypothetical beliefs we produce for given observed data, $p(\theta;x,y)$. Thus, we can now see that our choice of joint model corresponds to different choices for approximating these quantities. Assuming that the LLM distribution is used directly for the conditional as per §4, we thus have that

- the prior–likelihood pairing corresponds to using the approximations $p_{\text{true}}(y;x) \approx \int p(\theta)p_{\text{LLM}}(y;[\theta,x])d\theta$ and $p(\theta;x,y) \approx p(\theta)p_{\text{LLM}}(y;[\theta,x])/\int p(\theta)p_{\text{LLM}}(y;[\theta,x])d\theta$; and
- the data–estimation pairing corresponds to directly specifying a model for $p_{\text{true}}(y;x)$ and then using the approximation $p(\theta;x,y) \approx p_{\text{LLM}}(\theta;[x,y])$.

The appropriateness of each of these options, therefore, comes down to how faithful these approximations are respectively to the true data distribution, $p_{\text{true}}(y;x)$, and how we actually derive our belief distribution on $\theta$ in practice once we have seen the new data.

The former of these considerations is difficult to control for as we simply do not know the true response distribution and it is hard to say which approach will thus estimate it best (though we can refer to the discussion in §4 to determine which best matches our *beliefs* about the true response distribution). However, we do know upfront how we plan to derive our belief distribution on $\theta$ in practice, so we can use this to guide which joint model we formulate our EIG from. Namely, we observe that (a) using the prior–likelihood EIG pairing equates to assuming we will make a *Bayesian update to our beliefs* on $\theta$ using the likelihood $p_{\text{LLM}}(y;[\theta,x])$, and (b) using the data–estimation EIG pairing equates to assuming we will make an *in-context update to our beliefs* on $\theta$, as we are treating $p(\theta;x,y)$ as $p_{\text{LLM}}(\theta;[x,y])$.

Our preference between the pairings should therefore be guided in part by *how we plan to update the model in practice*. In particular, if we plan to make pure Bayesian updates, then the prior–likelihood formulation will tend to yield an EIG that is more faithful to our updating procedure, while if we only make simple in-context updates, the data–estimation formulation will tend to yield a more faithful EIG instead.

The update we use in practice, namely taking $p(\theta;x,y) = p_f(\theta;[x,y])$ as outlined in §3.1, can be seen as being somewhere between the in-context and Bayesian updating: we initially sample from $p_{\text{LLM}}(\theta;[x,y])$, but then perform filtering and other steps. The relative extent to which it resembles each will be problem–dependent and again be linked to how much we trust the LLM to capture uncertainty in the space of $\theta$ vs. $y$.

For the settings we consider, we expect $p_f(\theta;[x,y])$ to generally be better approximated by a Bayesian update than an in-context update, aligning with our decision to use the prior–likelihood formulation. The reasons for this are that a) the filtering often removes a large proportion of the generated samples, especially at later experiment turns, with $p_{\text{LLM}}(\theta;h_{t-1})$ not fully incorporating information from the history; b) the maintaining of the set of one consistent hypotheses from one turn to the next encourages a more Bayesian behavior, with samples persisting unless contradicted by a new likelihood term; and c) the typical premature overconfidence of $p_{\text{LLM}}(\theta;h_{t-1})$ to a small number of hypotheses means it is typically unrepresentative of our beliefs.

These theoretical benefits are perhaps secondary to the more practical benefits from the ease of constructing an appropriate model in the prior–likelihood formulation and avoiding direct uncertainty estimation in the space of $\theta$. Nonetheless, they help confirm that our choices have not induced unnecessary mismatch between the EIG formulation and our updating procedure.

The picture here can get a somewhat more complicated once we move into the sequential BED setting. Here, our ultimate aim is actually to minimize $H[p(\theta; h_T)]$ at some final future horizon $T$. Now, we only care about intermediary belief states $p(\theta; h_t)$ through their aid in future decision making toward the goal of minimizing the final entropy. Thus, even if we are working with in-context updates, it might be the case that $p(\theta; h_t)$ only starts to produce a meaningful entropy once we have seen enough data to sufficiently narrow down the possibilities on $\theta$. The optimal behavior in such settings would be to learn a policy that directly targets this final belief state instead of sequentially targeting the incremental EIGs. However, this will typically not be computationally feasible in practice and we instead need to resort to a myopic decision-making strategy. It might thus still be better to use the prior–likelihood formulation in such myopic decision making settings, even if we are sequentially updating our beliefs on $\theta$ through in-context updates, if this allows us to better guide the sequential decisions towards our final objective. The coherence of Bayesian updating means that the converse is unlikely to be true, so this provides further evidence towards using the prior–likelihood formulation.

## C  EXTENDED RELATED WORK

**Information-based question answering with LLMs**  Several recent works have (explicitly or implicitly) looked at information gathering with LLMs. Most of these can be framed in a BED setting, with a *deterministic likelihood* (Piriyakulkij et al., 2023; Hu et al., 2024; Kobalczyk et al., 2025; Cooper et al., 2025), and can be seen as variants of our Split baseline. Piriyakulkij et al. (2023) use a deterministic 0/1 answer likelihood $p(a|x, q)$ via the LLM to prune items from a pre-enumerated finite set given a candidate question $q$. The question is selected by minimizing expected posterior entropy. They model user preferences with a binary ground truth, which would not be applicable in preference-elicitation scenarios with nebulous user profiles. Similarly, Hu et al. (2024) use a deterministic likelihood to minimize entropy over a finite set $\Omega$ in a closed-world setting. Kobalczyk et al. (2025) target ambiguous task specifications in open-ended generation tasks by sampling a set of hypotheses (placing a uniform prior over them) and viewing each question as a deterministic partition over those samples, looking for questions that split the samples roughly evenly. Cooper et al. (2025) compute posterior entropy over a working set of top $k$ hypotheses (without filtering) through heuristic pruning.

Wang et al. (2025) avoid the pitfall of deterministic likelihoods. They use a data–estimation framework to estimate EIG, focusing on scenarios where the target can be expressed as a predefined series of multiple-choice questions. Their approach relies on meta-training a predictive language model on historic question/answer pairs, and so is not directly comparable with BED-LLM which requires no additional training or data. Chan et al. (2025) do not model likelihoods or posterior beliefs, instead they rely on the expected size of conformal prediction sets as a surrogate uncertainty metric. This requires the use of an additional calibration dataset, and is confined to closed-world settings with a finite label set and pre-defined queries.

**Post-training LLMs for improved information gathering**  Rather than augmenting a frozen LLM with the ability to estimate utility functions, some works have instead aimed to post-train an LLM to improve its ability to ask questions (Zhang et al., 2024; Wu et al., 2025; Andukuri et al., 2024). Most do not explicitly consider informativeness of questions: Zhang et al. (2024) and Wu et al. (2025) use reinforcement learning techniques to reward generations that quickly lead to the correct answer, and Andukuri et al. (2024) builds on Li et al. (2025b) by fine-tuning on successful traces. Mazzaccara et al. (2024) do indirectly incorporate uncertainty, also using a deterministic likelihood: they use predictive entropy to identify informative questions, and then either fine-tune on the highest-entropy question, or perform DPO comparing the highest-entropy question with a lower-entropy question. We do not address fine-tuning in this work, focusing instead on exploring the correct way to formulate BED using LLMs.

**Combining LLMs with parametric models**  As discussed in §4, a key challenge in adapting BED to the LLM setting is in aligning the expected information gain with the actual uncertainties extracted from the LLM after updating. Handa et al. (2024) take a different approach to this problem by using

---

**Algorithm 1** Data–Estimation sequential information gathering

---

**Require:** LLM $p$, history $h_0 = \varnothing$, budget $T$, num. hypothesis samples $M$

1: **for** $t = 1, \ldots, T$ **do**
2:     **Generate candidates:** sample $\mathcal{X}_{\text{cand}} = \{x_t^{(1)}, \ldots, x_t^{(K)}\}$ from $p(x_t; h_{t-1})$
3:     **for** each candidate $x_t \in \mathcal{X}_{\text{cand}}$ **do**
4:         **for** each $y \in \mathcal{Y}$ **do**                                 *// enumerate answer options*
5:             $w_y \leftarrow p(y; [h_{t-1}, x_t])$                      *// answer prob. via LLM logits*
6:             Sample $\{\theta_m^{(y)}\}_{m=1}^M \sim p(\theta; [h_{t-1}, x_t, y])$
7:             Estimate $\hat{H}_y \leftarrow \widehat{H}[p(\theta; [h_{t-1}, x_t, y])]$ using $\{\theta_m^{(y)}\}$
8:         **end for**
9:         $\hat{H}_{\text{cond}} \leftarrow \sum_{y \in \mathcal{Y}} w_y \hat{H}_y$                *// Term 1: marginal entropy of $\theta$ (design-dependent)*
10:         Pool: $\{\theta_m\}_{m=1}^{M'} \leftarrow \bigcup_{y \in \mathcal{Y}} \{\theta_m^{(y)}\}$                *// samples from mixture*
11:         **for** $m = 1, \ldots, M'$ **do**
12:             $\ell_m \leftarrow \log \sum_{y \in \mathcal{Y}} w_y \cdot p(\theta_m; [h_{t-1}, x_t, y])$
13:         **end for**
14:         $\hat{H}_{\text{marg}} \leftarrow -\frac{1}{M'} \sum_{m=1}^{M'} \ell_m$
15:         $\widehat{\text{EIG}}(x_t) \leftarrow \hat{H}_{\text{marg}} - \hat{H}_{\text{cond}}$
16:     **end for**
17:     **Select:** $x_t^* \leftarrow \arg\max_{x_t \in \mathcal{X}_{\text{cand}}} \widehat{\text{EIG}}(x_t)$
18:     **Ask and observe:** pose $x_t^*$, receive $y_t$, set $h_t \leftarrow (h_{t-1}, (x_t^*, y_t))$
19: **end for**

---

the LLM to generate features for an external conventional Bayesian joint model (in their case, a linear Bradley–Terry model), rather than deriving their joint model more directly from the LLM itself. This can be a good choice when the problem is well-bounded and we already have a well-specified Bayesian model form for the problem at hand; however, this may be challenging in arbitrarily large and complex hypothesis spaces. In particular, their specific method is not applicable more widely beyond the preference learning context they consider.

**BED** It has been noted that the traditional sequential BED approach can sometimes be suboptimal in practice, as it only optimizes the EIG of the next observation, without planning ahead for the fact that design decisions taken at a given step can also influence the achievable EIGs from future steps (Foster, 2021). A variety of *policy–based* BED approaches have subsequently been proposed to address this (Foster et al., 2021; Ivanova et al., 2021; Blau et al., 2022; Huan & Marzouk, 2016; Hedman et al., 2025), while also removing the need to make model updates and conduct optimizations during the experiment itself. Our findings are complementary: by providing more faithful model factorizations, belief updates, and EIG estimators in the LLM setting, BED-LLM could supply stronger building blocks for policy-based methods, reducing variance, enhancing effectiveness, and improving the sample efficiency of policy training.

## D   DATA–ESTIMATION METHOD

Our *Data–Estimation* method is based on a model derived from a data–estimation pairing (§3).

$$p(\theta, y_t; h_{t-1}, x_t) = p(y_t; [h_{t-1}, x_t]) \, p(\theta; [h_{t-1}, x_t, y_t]) . \tag{5}$$

### D.1   EIG ESTIMATION

Using the form of the EIG given by Eq. 1, we have

$$\text{EIG}_\theta(x) = \underbrace{H[p(\theta; h_{t-1}, x)]}_{\text{marginal entropy of } \theta} - \underbrace{\mathbb{E}_{p(y_t; [h_{t-1}, x_t])}[H[p(\theta; [h_{t-1}, x_t, y_t])]]}_{\text{expected posterior entropy}} . \tag{6}$$

Crucially, the marginal on $\theta$ implied by this pairing is obtained by integrating out $y$:

$$p(\theta; h_{t-1}, x) = \sum_{y_t} p(y_t; [h_{t-1}, x_t]) \, p(\theta; [h_{t-1}, x_t, y_t]) , \tag{7}$$

which is a mixture distribution whose components and weights both depend on $x$. Unlike in the prior–likelihood pairing, where the belief state $p_f(\theta; h_{t-1})$ is constructed independently of the candidate design, the marginal entropy $\mathrm{H}[p(\theta; h_{t-1}, x)]$ here *varies across candidate questions and cannot be dropped* from the optimization. This is a concrete manifestation of the issue identified in §3: the data–estimation pairing does not provide a design–independent belief state on $\theta$.

**Discrete answer space.** When the possible values for $y$ are discrete (and finite) and we can evaluate $p(y_t; [h_{t-1}, x_t])$ in closed form, both terms of Eq. 6 can be computed directly. The second term (expected posterior entropy) is:

$$\mathbb{E}_{p(y_t; [h_{t-1}, x_t])}[\mathrm{H}[p(\theta; [h_{t-1}, x_t, y_t])]] = \sum_{y_t} p(y_t; [h_{t-1}, x_t]) \,\mathrm{H}[p(\theta; [h_{t-1}, x_t, y_t])] \,, \qquad (8)$$

where each posterior entropy $\mathrm{H}[p(\theta; [h_{t-1}, x_t, y_t])]$ can be evaluated from the LLM's logits or, if these are unavailable, estimated by sampling. The first term (marginal entropy) requires evaluating $p(\theta; h_{t-1}, x)$ as the mixture in Eq. 7, then computing its entropy. In practice, this entropy cannot be obtained in closed form and must be estimated. A natural approach is to draw samples $\theta_m \sim p(\theta; h_{t-1}, x)$ (by first sampling $y_m \sim p(y_t; [h_{t-1}, x_t])$, then $\theta_m \sim p(\theta; [h_{t-1}, x_t, y_t]) [, y_m])$ and using a plug-in or nearest-neighbor entropy estimator, or evaluating $\log p(\theta_m; h_{t-1}, x) = \log \sum_y p(y_t; [h_{t-1}, x_t]) \, p(\theta_m; [h_{t-1}, x, y])$ exactly by enumerating over $y$.

**Monte Carlo estimator.** When $y$ cannot be enumerated, both terms must be estimated by Monte Carlo. Drawing $N$ samples $y_n \sim p(y_t; [h_{t-1}, x_t])$, the full estimator takes the form:

$$\mathrm{EIG}_\theta(x; h_{t-1}) \approx \mathrm{H}\left[\frac{1}{N}\sum_{n=1}^{N} p(\theta; [h_{t-1}, x, y_n])\right] - \frac{1}{N}\sum_{n=1}^{N} \mathrm{H}[p(\theta; [h_{t-1}, x, y_n])] \,. \qquad (9)$$

Whereas Eq. 3 performs this computation over the answer space $\mathcal{Y}$ (which is small and enumerable for multiple-choice questions), Eq. 9 requires it over the hypothesis space $\Theta$. For the first term, evaluating the mixture entropy requires computing $p(\theta; [h_{t-1}, x, y_n])$ for a set of sampled $\theta$ values across all $N$ mixture components — an $O(M \times N)$ cost in autoregressive probability evaluations over the full hypothesis text.

## D.2 IMPLEMENTATION

We provide an overview of the Data–Estimation procedure in Algorithm 1. In our experiments, the answer space $\mathcal{Y}$ is enumerable (multiple-choice options), so we use Eq. 8 for the second term. For the first term, we estimate the marginal entropy by sampling $\theta$ values from the mixture in Eq. 7 and evaluating $\log p(\theta; h_{t-1}, x)$ by enumerating over $y$; that is, we use:

$$\mathrm{H}[p(\theta; h_{t-1}, x)] \approx -\frac{1}{M}\sum_{m=1}^{M} \log\left(\sum_y p(y_t; [h_{t-1}, x_t]) \, p(\theta_m; [h_{t-1}, x, y])\right) \,, \qquad (10)$$

where $\theta_m$ are drawn from $p(\theta; h_{t-1}, x)$. This requires $M \times |\mathcal{Y}|$ evaluations of $p(\theta_m; [h_{t-1}, x, y])$ per candidate question, each involving a full forward pass over the hypothesis text.

## D.3 GENERATING CANDIDATE HYPOTHESES

To generate candidate values of $\theta$ for the data–estimation method, we use the prompt in Fig. 6.

Fig. 5 shows an example of the distribution of samples obtained following two rounds in the 20–questions game. Note that the samples are highly concentrated on just a handful of answers. This lack of diversity shows that the model's belief distribution is far more concentrated relative to the variability over valid hypotheses in the ground–truth task distribution, which negatively impacts the performance of the data–estimation method.

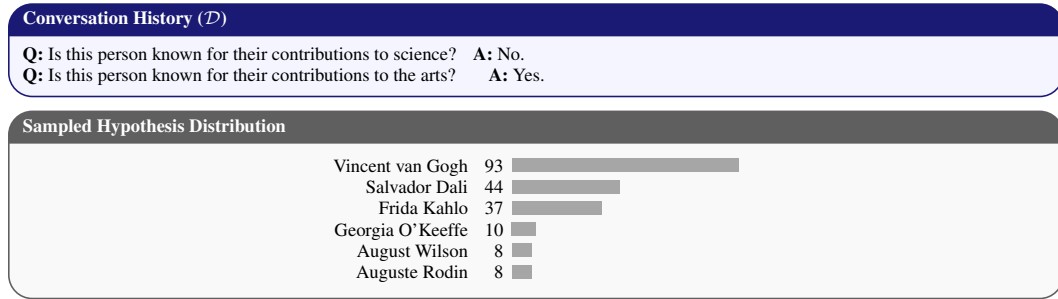

Figure 5: An empirical hypothesis distribution generated by the LLM, conditioned on the conversation history (top). By independently sampling $N = 200$ hypotheses, we observe significant *mode collapse*; the probability mass is heavily concentrated on a few prominent figures (e.g., Vincent van Gogh). This overconfidence negatively impacts the performance of the data-estimation baseline. Note that this aggregated view is purely diagnostic; Algorithm 1 operates directly on the probabilities of individual samples.

Figure 6: Prompt for generating hypotheses (and evaluating their probability) for the data–estimation method.

```
Return only the full name of one randomly selected famous person (living or deceased)
consistent with the questions and answers above.

To increase randomness:
1.  Internally brainstorm a pool of diverse and representative individuals.
2.  Avoid defaulting to the most globally ubiquitous celebrities or famous figures.

Output rules:
- Output ONLY the person's full name (with spaces, capitalization and accents), nothing
else.
- No extra words, explanations, numbering, or punctuation beyond what's in the name
itself (hyphens/apostrophes allowed if part of the name).
```

Fig. 7 shows an example of in-context belief updating failing to respect the conversation history. The response to the first question indicates that the target is male; however, after the second question, the LLM generates two female candidate hypotheses.

Figure 7: An example of in-context belief tracking with GPT-4o-mini proposing hypotheses inconsistent with the history.

```
Q1:  Is the person male?
Answerer:  Yes.

Q2:  Is this person often associated with civil rights or social justice?
Answerer:  Yes.

Sampled hypotheses:  ['James Baldwin', 'A. Philip Randolph', 'Angela Davis', 'Malcolm
X', 'W.E.B. Du Bois', 'Desmond Tutu', 'Cesar Chavez', 'Rosa Parks', 'Martin Luther
King Jr.', 'Langston Hughes', 'John Lewis', 'Frederick Douglass', 'Nelson Mandela',
'Thurgood Marshall', 'Bayard Rustin']

Hypotheses rejected with filtering:  ['Angela Davis', 'Rosa Parks']
```

# E    GENERATING CANDIDATE HYPOTHESES FOR BED-LLM

Figure 8: Prompt for generating candidate hypotheses for the "Things" dataset. Similar prompts were used for "Celebrities" and "Animals".

```
You are playing a game of 20 Questions.  Using all of the questions and answers so far:

Generate up to {num_samples} candidate entities that satisfy every clue.
Each candidate must be a single, self-contained entity (e.g., "Europa", "Bagpipe",
"Diadem").
List each entity on its own line - no numbering, punctuation, or extra text.
Produce a varied set by identifying features not implied by the clues and diversifying
along them.
Do not repeat any entity.

Return only the list of entities.
```

A fundamental challenge for BED-LLM and its ablations is generating a sufficiently diverse set of candidate hypotheses from the LLM's belief distribution, that are consistent with the previously-answered questions. Below, we detail the steps we take to construct our distribution over hypotheses.

**Candidate hypotheses are generated jointly, rather than independently**    As illustrated in Fig. 5, the raw distribution $p_{\text{LLM}}(\theta)$ is highly overconfident, often concentrating mass on only a few high-likelihood hypotheses. Thus, it is not practical to directly use the LLM's distribution as a prior $p(\theta)$. Instead, we jointly sample candidates $\theta$ and assume a uniform distribution over them. We can view this as sampling $\theta$s from a mixture distribution. The LLM is prompted to generate a list of $N$ hypotheses in a single rollout, which corresponds to drawing from the autoregressive list distribution

$$p_{\text{LLM}}(\theta_t^{(1)}, \ldots, \theta_t^{(N)}; h_t) \;=\; \prod_{n=1}^{N} p_{\text{LLM}}(\theta_t^{(n)}; [\theta_t^{(1:n-1)}, h_t]) \,.$$

**We use a diversity-encouraging prompt**    We use a prompt designed to elicit stratified hypotheses by encouraging the LLM to consider different semantic features (e.g. age groups, genres, or categories) and implicitly diversify across them. An example prompt is shown in Fig. 8. In our generation prompt, we reverse the order of the question–answer pairs in $h_t$ to place the most recent question at the top of the context window (while retaining earlier exchanges), ensuring that specific constraints are prioritized and mitigating context drift. For the 20 Questions experiments, we used a higher-than-normal temperature ($T = 1.3$) to increase diversity of responses. For the preference elicitation experiments, we used $T = 1$ to obtain more coherent responses.

**Candidates are filtered based on the history**    For each candidate, we use $p_{\text{LLM}}(\theta; h_{t-1})$ to assess whether it is compatible with the previous question/answer pairs. We filter responses where the probability of the given answer falls below a certain threshold; in our experiments we set this threshold to 0.2.

**Valid candidates from previous generations are included**    We also filter the candidate hypotheses from the previous generation, based on the most recent question/answer pair, and include these in our candidate set. We repeat the generation process, keeping the previously generated and filtered samples in context to elicit new generations, either twice or three times if sufficient hypotheses have not been generated (noting the number of possible valid samples can be less than the number requested).

**We define a uniform prior over hypotheses**    While one could in principle reweight candidates using importance sampling, in practice we choose to not rely on the model's internal probabilities. Instead we define the prior as a uniform distribution:

$$p_f(\theta; h_t) \approx \frac{1}{|\Theta|} \sum_{\theta \in \Theta} \delta_\theta.$$

Finally, we note that different LLMs respond differently to strategies aiming to increase diversity: some benefit more from a higher temperature while others benefit from more repetitions of the sampling–filtering cycle. For fairness, in our experiments we have kept these parameters constant across models.

# F    ADDITIONAL EXPERIMENTAL RESULTS

## F.1    20 QUESTIONS AND PREFERENCE ELICITATION

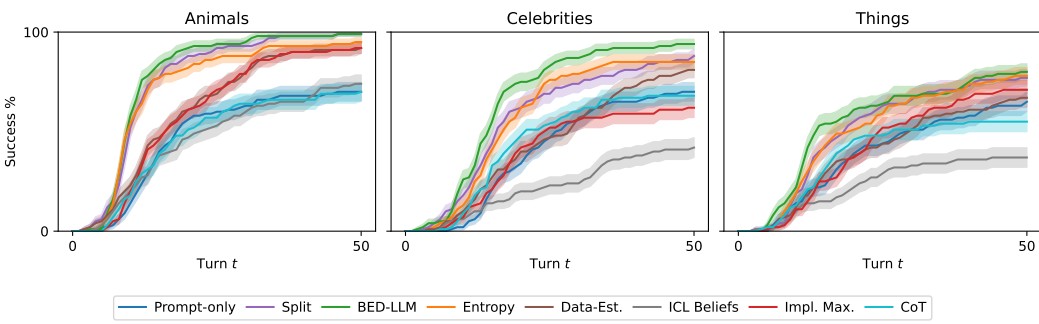

Figure 9: Success rate on 20 Questions with Qwen2.5-72B, extended to 50 turns: mean ± standard error across 100 targets per dataset. BED-LLM remains the best-performing method at every turn. The performance gap to baselines narrows in later turns as the task saturates — most methods eventually solve the easier instances — but BED-LLM consistently retains its advantage.

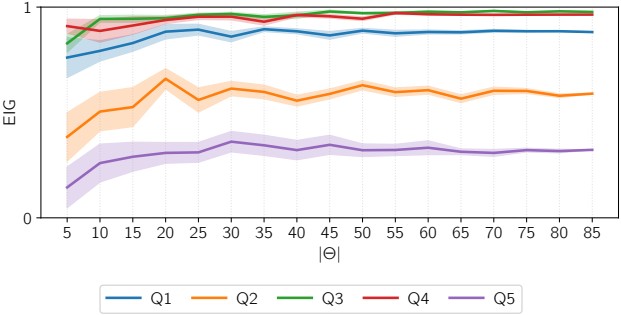

Figure 10: Analysis of the empirical convergence of the Rao-Blackwellized EIG estimator using GPT-4o-mini. We plot EIG estimates for 5 candidate questions at turn 5 of the 20 Questions game with $\theta^* =$"Saoirse Ronan", as a function of the number of samples in the hypothesis set $\Theta$.

Fig. 10 shows the EIG estimates for five candidate questions as the number of hypothesis samples $|\Theta|$ increases, at turn 5 of a representative 20 Questions game. The estimates converge rapidly with the number of samples, and more importantly, the *ranking* of candidate questions stabilizes well before the estimates themselves have fully converged. By $|\Theta| = 10$ the relative ordering of all five candidates already matches the ordering observed at $|\Theta| = 85$. This means that the question selection decision—which is what matters algorithmically, since we only need the $\arg\max$, not the absolute EIG value—is robust to using a small hypothesis set. This justifies BED-LLM's use of a moderate sample budget without sacrificing decision quality.

## F.2    WALL-CLOCK TIMES

Tab. 3 shows that Data–Estimation is over an order of magnitude slower than all other methods: its full-sequence likelihood evaluations per candidate question dominate the cost. Among the remaining methods, BED-LLM's runtime is comparable to—and even slightly lower than—Entropy and Split, despite computing the full EIG; this is because the additional conditional-entropy term in Eq. 3 reuses the same likelihood evaluations and adds negligible overhead. Prompt-Only and CoT are the cheapest methods, as they require only a single LLM generation call per turn with no hypothesis sampling or objective computation. Implicit Maximization falls in between: it samples hypotheses and candidate questions but replaces explicit EIG estimation with a single LLM selection call. ICL Beliefs is slightly more expensive than BED-LLM because the absence of filtering leads to lower-quality hypotheses, requiring a much larger number of questions before the game terminates.

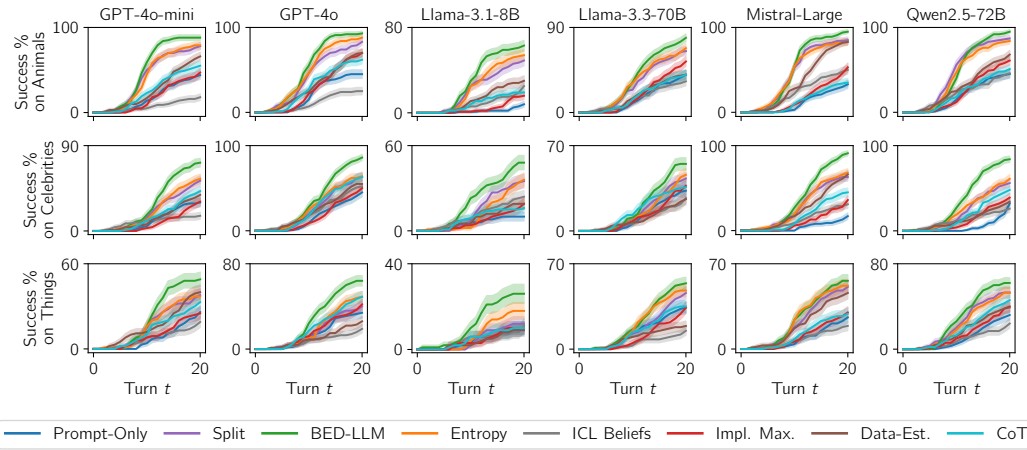

Figure 11: Success rate on 20 Questions. Mean $\pm$ standard error across 100 targets. BED-LLM beats all other methods across all datasets and models evaluated.

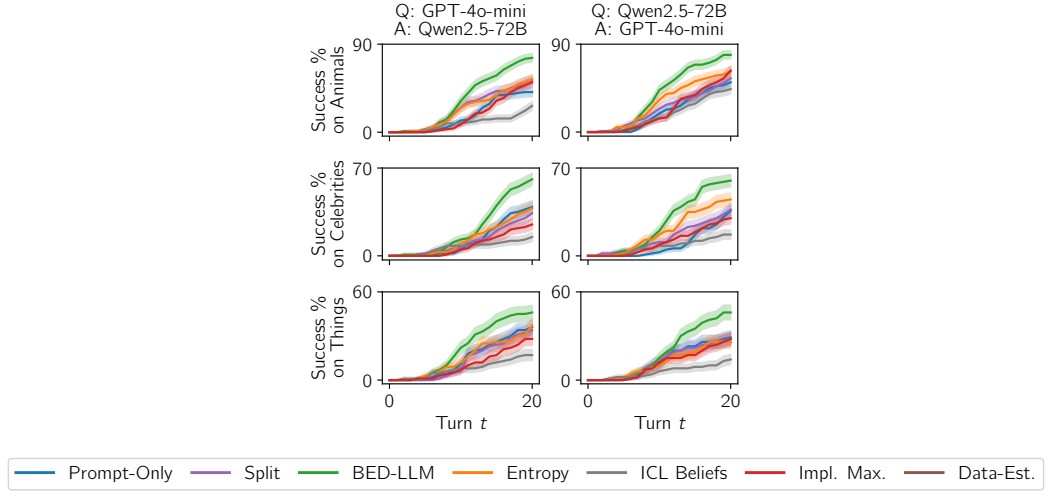

Figure 12: Full plots for 20 Questions Experiments with different questioner and answerer.

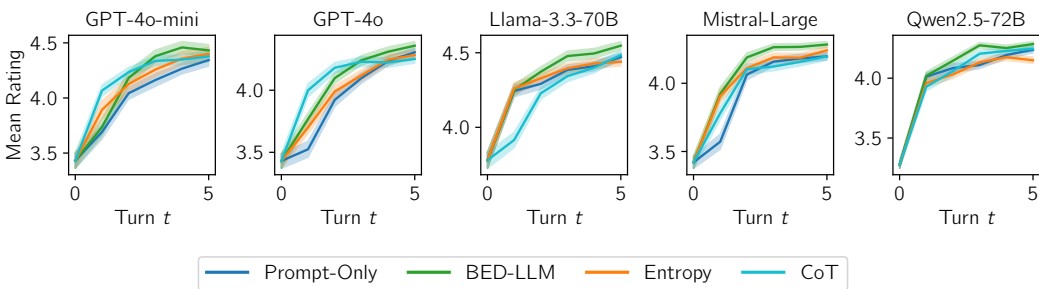

Figure 13: Mean rating across 10 film recommendations: mean $\pm$ standard error across 200 users. BED-LLM beats all other methods across all datasets and models evaluated.

| Method | Runtime |
|---|---|
| Data–Estimation | 1d 17h 13m 34s |
| ICL Beliefs | 3h 10m 39s |
| Entropy | 2h 46m 19s |
| Split | 2h 30m 35s |
| BED-LLM | 2h 28m 43s |
| Implicit Maximization | 45m 10s |
| CoT | 35m 28s |
| Prompt-Only | 26m 29s |

Table 3: Wall-clock runtimes for 20 Questions for Qwen2.5-72B on the entire *Animals* problem set.

## F.3 EFFECT OF SEQUENTIALLY UPDATING THE BED-LLM LIKELIHOOD

As discussed in App. F.3, we choose to use a *static* likelihood, $p_{\text{LLM}}(y_t; [\theta, x_t])$, in the model factorization for BED-LLM. Alternatively, we could condition the likelihood on the full interaction history, i.e. $p_{\text{LLM}}(y_{t+1}; [h_{t-1}, \theta, x_{t+1}])$. However, this can lead to undesirable effects (e.g. context-induced calibration shifts). We provide results in Tab. 4 for the 20 Questions game where we update the likelihood, and demonstrate that it leads to consistently worse performance than BED-LLM, supporting our rationale in App. F.3 to keep the likelihood static.

| Dataset | Model | Success Rate (%) | |
|---|---|---|---|
| | | **BED-LLM + static likelihood** | **BED-LLM + likelihood updating** |
| Animals | GPT-4o-mini | $88 \pm 3.3$ | $85 \pm 3.6$ |
| | GPT-4o | $93 \pm 2.6$ | $86 \pm 3.5$ |
| | Llama-3.1-8B | $63 \pm 4.9$ | $67 \pm 4.7$ |
| | Llama-3.3-70B | $79 \pm 4.1$ | $80 \pm 4.0$ |
| | Qwen2.5-72B | $95 \pm 2.2$ | $89 \pm 3.1$ |
| Celebrities | GPT-4o-mini | $72 \pm 4.5$ | $63 \pm 4.9$ |
| | GPT-4o | $86 \pm 3.5$ | $83 \pm 3.8$ |
| | Llama-3.1-8B | $58 \pm 5.0$ | $56 \pm 5.0$ |
| | Llama-3.3-70B | $55 \pm 5.0$ | $44 \pm 5.0$ |
| | Qwen2.5-72B | $84 \pm 3.7$ | $70 \pm 4.6$ |
| Things | GPT-4o-mini | $49 \pm 5.0$ | $43 \pm 5.0$ |
| | GPT-4o | $64 \pm 4.8$ | $57 \pm 5.0$ |
| | Llama-3.1-8B | $26 \pm 4.4$ | $24 \pm 4.3$ |
| | Llama-3.3-70B | $55 \pm 5.0$ | $54 \pm 5.0$ |
| | Qwen2.5-72B | $62 \pm 4.9$ | $61 \pm 4.9$ |

Table 4: BED-LLM success rates across datasets and models. $\pm$ numbers show the standard error of the mean estimated using $\sqrt{p(1-p)/(n-1)}$ where $p$ is the success percentage and $n$ is the number of datapoints. This estimator is positively biased and thus conservative.

## G FURTHER DESCRIPTION OF METHODS AND ABLATIONS

**Prompt-Only**: A direct prompting baseline where the model is asked to immediately generate the next question given the dialogue history, no candidate generation, and no belief modeling (Fig. 14). This corresponds to the simplest and most common way users currently interact with LLM agents for question-asking tasks, and therefore serves as a natural "standard LLM prompting" baseline. After the LLM has generated its question, it waits for the user response and generates a new question based on the new history.

---

**Figure 14: Prompt for the Prompt-Only method.**

```
Your task is to ask a series of questions to deduce the identity of the 'person'
if task == 'celebrities' else 'animal' if task == 'animals' else 'entity' that I'm
thinking of with as few queries as possible.
Only ask questions that can be answered by 'yes', 'no' or 'maybe'.
Do not ask for hint.  Make your question standalone with no linebreaker.

Output format:  <question>Your question here?</question>

Now start asking a question.
```

---

**CoT**: A direct prompting baseline where the model is asked to reason and provide action (Fig. 15) loosely following the ReAct framework (Yao et al., 2023b).

---

**Figure 15: Prompt for the CoT method.**

```
Your task is to ask a series of questions to deduce the identity of the 'person'
if task == 'celebrities' else 'animal' if task == 'animals' else 'entity' that I'm
thinking of with as few queries as possible.
Only ask questions that can be answered by 'yes', 'no' or 'maybe'.
Do not ask for hints.  The actual question you ask must be standalone with no
linebreakers.

Use the following format for every response:
Thought:  <brief reasoning about what to ask next>
Action:  ASK["<a single yes/no/maybe question with no linebreaks>"]
Now start by producing your first Thought and Action.
```

---

**Split**: Essentially a version of our method in which we cast previous approaches, such as Kobalczyk et al. (2025): uses the same candidate question generating method as BED-LLM, assumes a deterministic likelihood over hypotheses (i.e., each hypothesis deterministically predicts a single answer to each question), and Evaluates the question on the sampled hypotheses, and selects the question that maximally balances the hypotheses, i.e., splits the current hypothesis set into subsets whose sizes are as close to equal as possible.

**Implicit Maximization (IM).**   The *Implicit Maximization* baseline is a lightweight, reasoning-driven method inspired by the *Tree-of-Thoughts* (ToT, Yao et al., 2023a) framework. ToT methods explicitly expand a search tree of intermediate reasoning steps ("thoughts") before selecting an action. Implicit Maximization provides a computationally efficient, collapsed variant of this idea:

- Rather than explicitly branching over thoughts, the model is prompted to internally deliberate over several possible next questions and their consequences.
- The branching and evaluation occur inside the model's chain-of-thought, rather than through externally enumerated tree expansion.
- The model then outputs the question it judges to be most informative, performing an amortized, ToT-style lookahead within a single LLM call.

Thus, IM captures the central intuition of Tree-of-Thoughts—reasoning over hypothetical futures—while avoiding the substantial computational cost of explicit tree search. It serves as a strong inference-time reasoning comparator to BED-LLM.

## H    EXPERIMENT DETAILS FOR 20 QUESTIONS

### H.1    PROBLEM SETS

We evaluate across three distinct problem sets—Animals, Celebrities, or Things—with each containing a mix of 100 obscure and common targets. Here, the problem set is just a list of different $\theta^*$ that will be individually provided to the answerer to instantiate different problems (e.g. we conduct a trial where $\theta^* =$ "dog", then one where $\theta^* =$ "cat", etc). The list of targets is *never* provided to the questioner model to restrict the set of possible hypotheses: the questioner is only prompted that is trying to identify an "animal", "celebrity", or "thing". The problem sets are

- Animals: a set of animal species generated with OpenAI's o3 model to ensure a diverse mix and balanced taxonomy.

- Celebrities: a diverse set of public figures, as used by Zhang et al. (2024).

- Things: a collection of everyday and exotic entities drawn from the web corpus, as used by Zhang et al. (2024); it covers a wide range of categories, from plants and clothing to professions, events, and mythical creatures.

To create the Animals problem set, we prompted OpenAI o3 (OpenAI, 2025, o3-2025-04-16) to generate a list of animals, using the prompt in Fig. 16. The resulting list is shown in Fig. 17. Alternative names (after | character) were manually added.

---

**Figure 16: Prompt for Animals problem set generation.**

```
You are a zoologist.
Please generate a list of 100 living animal species with very high taxonomic diversity,
including diversity in phyla, classes, orders, and families.  Present each animal on a
different line.
```

African elephant
Bengal tiger
Bald eagle
Blue whale
Red kangaroo
Giant panda
Snow leopard
Green sea turtle
American alligator
Bottlenose dolphin
Emperor penguin
Great white shark
Golden poison frog | Golden poison dart frog
Honey bee
Monarch butterfly
Okapi
Chimpanzee
Arctic fox
Komodo dragon
Giraffe
Cheetah
Hammerhead shark
Axolotl
Orca
Puffin
Red panda
Platypus
Rhinoceros beetle
Tasmanian devil
Wombat
Sloth
Blue-ringed octopus | Blue ringed octopus
Manatee
Narwhal

Sea otter
Coral snake
King cobra
Harpy eagle
Lemur
Koala
Aye-aye | Ayeaye
Snowy owl
Elk
Wolverine
Caracal
Cassowary
Quokka
Pangolin
Saiga antelope
Galápagos tortoise | Galapagos tortoise
Sumatran orangutan
Red-eyed tree frog | Redeyed tree frog
European badger
Moose
African grey parrot
Scarlet macaw
Black mamba
Albatross
Humpback whale
Dugong
Anaconda
Kookaburra
Coyote
Brown bear
Golden jackal
Capybara
Ibex
Japanese macaque

Kiwi
Leafcutter ant
Mantis shrimp
Ocelot
Peregrine falcon
Quetzal
Raccoon
Sand cat
Tarantula
Uakari
Vicuña
Wildebeest
Rock hyrax | dassie
Yak
Zebra
Blue dragon nudibranch | Blue dragon sea slug
Chinchilla
Dhole
Electric eel
Flying fox
Gharial
Horseshoe crab
Indigo bunting
Jerboa
Kakapo
Lionfish
Markhor
Nautilus
Olive baboon
Pika
Quoll
Rosy boa

Figure 17: Animals problem set (generated using OpenAI o3, with manual curation)

## H.2 EVALUATION

We assess performance by tracking the questioner's ability to identify the hidden target $\theta^*$ over the course of each game. At each turn $t$, we prompt it to produce a single guess for $\theta^*$ via greedy decoding—that is, we extract the highest likelihood candidate from the belief state of the questioner $p_f(\theta; h_t)$. This guess is evaluated against the true target $\theta^*$ (including alternative names) using case–insensitive exact string matching and we measure the proportion of correct guesses at each turn. Importantly, these evaluation guesses are *not* part of the questioner algorithm itself: they are extractions from the questioner's belief state $p_f(\theta; h_t)$ and are excluded from $h_{t-1}$ to not affect subsequent question selection. In line with the original rules of the game, we also introduce an explicit mechanism for the questioner to guess the answer as part of its 20 questions: if the set of filtered hypotheses collapses to a single candidate, or a direct guess of $\theta^*$ is evaluated as the maximally informative question by the acquisition function, the questioner asks "Is it ⟨item⟩?". This guess is evaluated using exact string matching, as above. If there is a match, the game terminates successfully; otherwise, if $t < 20$, the game continues with the question and negative response included in $h_{t-1}$ and counted towards the 20 question budget.

## H.3 ALGORITHMIC DETAILS

Using our sample–then–filter process (see §3.1), we aim to sample at least $N = 15$ hypotheses, repeating the cycle up to three times if needed (the exact number of hypotheses can be less than this as it may not be possible to generate sufficient valid hypotheses, especially in later experiment turns). The questioner generates $M = 15$ candidate questions to test, $\mathcal{X}^{\text{cand}}$, using the "conditional generation" approach of §3.2 when possible, but falling back on "unconditional generation" if insufficient candidate hypotheses have been generated.

# I EXPERIMENT DETAILS FOR PREFERENCE ELICITATION

## I.1 PROBLEMS

To generate a set of ground-truth user profiles, we take a set of 200 real user ratings from the MovieLens-100K dataset (Harper & Konstan, 2015), then use an "oracle" LLM (namely, OpenAI's o3 model) to produce a paragraph of text that is consistent with each distinct set of ratings, using the prompt in Fig. 18. As was the case for the 20 Questions problems, this problem set is never provided to the questioner and the set of allowed $\theta$ is not constrained.

Because we need the LLM to be able to meaningfully capture uncertainty in the space of responses, we restrict questions to be multiple-choice. Specifically, the questioner is tasked with producing a question along with five possible responses A/B/C/D/E. We then define each $x_t$ to be the question coupled with the possible responses, and each $y_t$ to be one of the letters A/B/C/D/E to provide a restricted set of tokens over which we can measure entropy. Option E is further constrained to always be "none of the above" so that the answerer is not committed to choosing one of the directly generated choices if none are suitable.

## I.2 EVALUATION

To again allow tracking of progress through the experiment, after each turn of the interaction $t$, the questioner generates ten film recommendations, conditioned on $h_{t-1}$. These recommendations are checked for consistency with prior questions and answers; if any are judged inconsistent then they are removed and additional recommendations are generated. The quality of the film recommendations is then assessed using an "LLM-as-judge" protocol (Zhu et al., 2025; Trivedi et al., 2024). Namely, the answerer evaluates each of the 10 films recommended by the questioner, conditioned on the hidden target user profile $\theta^*$. It scores each film on a scale of 1 to 5 (in 0.5 increments), based on how well the recommendation aligns with $\theta^*$ — this score is output together with a brief justification to increase reliability. We report the mean rating and standard error across 200 users, over 5 question–answer turns.

### I.3 ALGORITHMIC DETAILS

For BED-LLM and Entropy, we compare $M = 8$ candidate questions at each turn and we aim to generate at least $N = 5$ candidate hypotheses. We use the "unconstrained generation" approach of candidate question generation (see §3.2) as the user profiles can be quite diffuse and we are only generating a small number of possible hypotheses that can be quite easy to split.

We note that data–estimation setup is not at all viable for this problem because the large number of tokens and varying dimensionality of each $\theta$ sample mean that $\text{H}[p_{\text{LLM}}(\theta; [h_{t-1}, x_{t+1}, y_{t+1}])]$ is not only infeasible to estimate, but also is not a meaningful measure of uncertainty.

Figure 18: Prompt used to generate ground-truth user profiles for preference elicitation task.

```
You will be given a user's complete film rating history from the MovieLens dataset,
provided as a dictionary structured by rating levels.
Your task is to thoroughly analyze the user's preferences across the entire range of
their film ratings (from highest to lowest).  Then, write a cohesive, descriptive
paragraph (approximately 5-7 sentences) summarizing the user's overall film taste
profile.

In your response, explicitly address:

Favored Elements (inferred primarily from 4.5-5.0 ratings):

        • Highlight the genres, narrative styles, themes, tones, historical eras, and
          emotional experiences that consistently resonate positively with this user.

        • Avoid mentioning any specific film titles, characters, or explicit plot points.

Neutral or Mixed Preferences (inferred from ratings around 2.5-4.0):

        • Note if there are indications of genre overlap or conditional enjoyment,
          such as certain genres or styles they occasionally appreciate under specific
          circumstances.

Disliked Elements (inferred primarily from 0.5-1.5 ratings):

        • Clearly outline the genres, narrative characteristics, tones, or emotional
          impacts that the user consistently finds unappealing or poorly executed.

Your paragraph must be precise, informative, nuanced, and balanced, effectively
capturing the complexity and specificity of the user's movie preferences.  The
resulting profile should be clear and detailed enough for a recommendation system
to accurately predict the user's likely enjoyment or dislike of other films based on
their established patterns of taste.

Proceed carefully, reasoning explicitly about the user's overall rating patterns rather
than relying exclusively on extreme ratings, to form a comprehensive, stable, and
representative film preference profile.
```

## J CODE AVAILABILITY

Code is available at https://github.com/DeeproChoudhury/BED-LLM.

