# OpenReview forum: "BED-LLM: Intelligent Information Gathering with LLMs and Bayesian Experimental Design"
_ICLR.cc/2026/Conference — ICLR 2026 Poster_

### Official Review · Reviewer_4sXe · 2025-10-30

**Soundness:** 2
**Presentation:** 2
**Contribution:** 2
**Rating:** 4
**Confidence:** 3

**Summary:**

The paper introduces a Bayesian Experimental Design (BED) inspired algorithm that enables LLMs to ask informed questions to a user or an external source. The approach leverages the internal belief probabilities of the LLM to estimate the prior and conditional distributions over queries and answers, allowing it to select optimal questions based on Expected Information Gain (EIG). EIG is computed as the difference in entropy between the posterior and prior beliefs, capturing the expected reduction in uncertainty after conditioning on the queried question and its potential answers.The algorithm proposes several strategies to construct and update the belief distributions through prompting and sampling from the underlying LLM. Notably, it uses an external LLM to generate potential questions candidates rather than a fixed set of known questions. Experiments on both the “20 Questions” and a preference elicitation benchmark show modest improvements compared to “Naive” and “Split” baselines.

**Strengths:**

1. The paper addresses an important challenge in information seeking for LLM, aiming to formalize how LLMs can ask meaningful and informative questions.
2. The proposed framework, based on Bayesian Experimental Design (BED) and Expected Information Gain (EIG), is theoretically well-motivated and offers strong interpretability and soundness for modeling information-seeking behavior.
3. The authors provide detailed explanations of their assumptions and offer clear justifications for these assumptions

**Weaknesses:**

The paper would benefit from improvements in clarity and presentation, which would enhance its accessibility to readers.

1. The overall structure of the paper is not well organized and is difficult to follow. The methodology section is dense and provides minimal examples to clarify key ideas. The authors should consider adding illustrative visuals or concrete examples to make the approach more understandable.

Some claims and assumptions are insufficiently motivated/addressed, and the arguments supporting them appear insufficient or not well justified. The author should refrain from making claims and general conclusion without strong supporting evidence.

2. Several claims in the paper are not sufficiently supported by experiments or reasoning. For instance, the authors state that the ``approach often fails to appropriately incorporate the information from $h_{t-1}$, leading to a belief distribution inconsistent with past observations,'' yet no examples, analyses, or ablations are provided to substantiate this claim.
3. Some simplified assumptions are concerning. In particular, replacing $p(y \mid \theta, x)$ with $p_{\text{LLM}}(y; \theta, x)$ when constructing the joint distribution $p(y, \theta; x)$ raises concerns about correctness and validity. In appendix A.1 the author's justification for not updating likelihood $p_{\text{LLM}}(y; [\theta, x_t])$  is "$\theta$ capture all the required information to predict y|x" which also lacks evidence to support the claim.

The paper lacks sufficient specification of experimental details and design choices, and provides limited baseline comparisons or ablation analyses. The two baselines included in the paper are not representative of existing approaches, offering little context for interpreting the reported results. Moreover, since the proposed method requires multiple LLM calls to generate candidate hypotheses and questions, reporting computational costs such as token usage or inference time would be essential to contextualize performance. Finally, the lack of detailed descriptions or illustrative examples for the baselines is concerning make it difficult to assess the fairness of the comparisons.

4. The paper does not discuss computational costs such as token usage, inference time, or scalability. Including such details would provide important context for assessing the practicality of this approach beyond toy problems.
5. Prompt based methods such as Reflexion, ReAct, and CoT should have been included for a more meaningful evaluation.
6. The experimental setup is not clearly explained. For example, the paper does not define what the `"Naive'' baseline represents, and the "Split'' baseline is only vaguely described. The comparison to standard LLM prompting methods is also limited---For the preference elicitation task, including only the ``Naive'' baseline is insufficient.
7. The paper lacks ablation studies analyzing the impact of different design choices. Without these analyses, it is difficult to determine which components of the proposed approach actually contribute to the reported performance improvements.

I think the paper is well motivated and the core idea is well-justified. The paper itself requires substantial improvement in clarity, organization, and experimental rigor. The presentation is difficult to follow, several design choices and assumptions are not well motivated, and key claims lack sufficient empirical support or ablation analysis.

**Questions:**

1. The paper uses $p_{\text{LLM}}(y; \theta, x)$ as an approximation of $p(y \mid \theta, x)$. Is there evidence demonstrating that this approximation is sound? Specifically, have you provided empirical validation from your own experiments or cited related work that supports this substitution? A clearer justification seems necessary, given that this approximation is central to the computation of the joint probability $p(y, \theta, x)$.

2. In line 159, you state that "autoregressive sequential rollouts often lead to more nuanced and diverse behavior than repeated static likelihood queries.'' Could you elaborate on why having ``more nuanced and diverse behavior'' is important in a BED framework? What advantage does this bring in relation to your work?

3.  In Appendix~A.1, you mention that "we choose not to update the likelihood model as more data is gathered,'' meaning $h_{t-1}$ is not used to update the likelihood. The justification provided---"for many problems, our beliefs on $\theta$ capture all the required information to predict $y|x$''---seems inconsistent with your prior, which does depend on $h_{t-1}$. Wouldn’t it therefore be more accurate to include $h_{t-1}$ in your experiments rather than exclude it? I’m not fully convinced by the argument that doing so is "not necessary.''

4.  The paper claims that one advantage of the proposed method is that, unlike prior work, it is not restricted by pre-specifying allowable entities: "In general, these previous works have also required restrictions on the space of allowable hypotheses $\theta$, whereas we do not tell the LLM this set of target entities, so the space of $\theta$ is bounded only by what the LLM can generate.'' However, your approach still requires $\theta$ when computing the EIG. Is the key difference that prior work assumes $\theta$ is explicitly given, while your approach allows $\theta$ to be generated by the LLM?

5. Could you provide details on how the "Naive'' and "Split'' baselines are implemented? The current manuscript lacks sufficient description of their setup. Additionally, how does the number of possible answers (20 vs. 500) affect performance? Does it influence your results?

---

> ### Author Response · Authors · 2025-11-21
>
> Thank you for your detailed review and helpful feedback.  We hope the below and our updated paper address the concerns you raised.
>
> **Summary**: “[Experiments] show modest improvements compared to the Naive and Split baselines”
> We are a little confused by this assertion that the gains are modest. Our BED-LLM approach outperformed all baselines across all experiments and the gains over these baselines are often huge as shown in Table 1, often doubling the success rate of the Naive method.  For example, on 20 questions the average success rates on Animals were 36.4% for Naive, 72.4% for Split, and 83.6% for BED-LLM; on Celebrities were 30% for Naive, 50% for Split, and 71% for BED-LLM; and on Things were 27.2% for Naive, 37.4% for Split, and 51.2% for BED-LLM.
> Thus, overall, we are on average outperforming Naive by 37.4 percentage points and Split by 15.3 percentage points. These are far bigger gains than typically achieved in machine learning papers over previous state-of-the-art.
> The gains on preference elicitation are less dramatic because the baselines already do relatively well, but we still see consistent improvements across the board.
> It is also notable here that the Split baseline represents what was previously considered state-of-the-art, but is a strictly inferior approach to the Entropy baseline we compared to (as Split can be viewed as a numerical approximation of the Entropy approach) and Entropy actually loses to Naive on 4 of the 6 tasks, so the consistency of BED-LLMs gains here is really quite notable.
>
> **W1 (structure)**. Thank you for the suggestions. Our current draft already contains several illustrative cases:
> - Appendix D: an example showing how the LLM’s implicit belief state can over-collapse when updated purely via in-context summaries, without our diversity-oriented sampling strategies (co-sampling and filtering) which prevent this issue. Extended details of our diversity sampling and how we carry out filtering to resolve inconsistencies with past observations is detailed in Appendix E.
> - Appendix A.2: an example showing how using entropy or uncertainty-sampling can select questions that the model is unsure about but are low-information, highlighting the need for full EIG.
>
> Based on your feedback, we plan to use the extra page allowed for the camera-ready version to move these to the main paper and also provide an additional worked example to help clarify the key ideas.  We have also added additional concrete examples of the BED-LLM method to Appendix H to clarify the intuition behind the belief update mechanism, EIG computation, and question-selection process. We believe these changes will make the structure easier to follow while preserving the technical rigor.
>
> **W2 (support for claims about ICL)**. We actually provide both an ablation and example to substantiate this claim, and have added an additional example. Firstly, the ICL beliefs baseline is exactly an ablation to show this failure. In this ablation, we replace the filtering beliefs $p_f(\theta; h_t)$ with pure LLM in-context beliefs, leading to a decrease in performance.  In some cases, "ICL beliefs” performs worse than the Naive baseline, demonstrating how strong the effect is.
>
> Secondly, Figure 5 in Appendix D.2 shows a qualitative example of a failure mode of ICL belief updating, whereby the distribution becomes overly collapsed – something our sampling procedure is designed to avoid.
>
> We added a qualitative example (see Figure 6) that shows how ICL can ignore information from previous responses, necessitating a filtering step. We reproduce this example below:
> ```
> Q1:  Is the person male?
> Answerer:  Yes.
> Q2:  Is this person often associated with civil rights or social justice?
> Answerer:  Yes.
>
> Sampled hypotheses:  ['James Baldwin', 'A. Philip Randolph', 'Angela Davis', 'Malcolm X', 'W.E.B. Du Bois', 'Desmond Tutu', 'Cesar Chavez', 'Rosa Parks', 'Martin Luther King Jr.', 'Langston Hughes', 'John Lewis', 'Frederick Douglass', 'Nelson Mandela', 'Thurgood Marshall', 'Bayard Rustin']
> Hypotheses rejected with filtering:  ['Angela Davis', 'Rosa Parks']
> ```
>
> It should be noticed that this failure mode is well-documented in prior LLM reasoning literature (e.g. chain-of-thought drift, context inconsistency, Liu et al 2024).
>
> More generally, we respectively disagree that the paper makes any unsupported claims, but we are happy to add any further evidence that you think is needed.

---

> > ### Author Response · Authors · 2025-11-21
> >
> > **W3 (assumptions)**. We believe there may have been a  misunderstanding of our approach here as we are not making any simplifying assumptions. In particular, we are not “replacing” some ground truth $p(y|\theta, x)$, we are selecting a likelihood that we believe is a good fit for modelling the responder; this is a core premise for all probabilistic machine learning (e.g. using Gaussian or Poisson likelihood isn’t the “true” distribution, it’s a modeling choice).  There is also no real viable alternative here unless we take a fully prompting based strategy or instead assume some model for $p(\theta|x,y)$ directly instead (yielding a variant of the data-estimation setup), both of which we show perform significantly worse empirically than our approach.  In fact, our approach is distinguished from related works in the area by _not_ making additional simplifying assumptions, like a deterministic likelihood, a restricted hypothesis space, or or relying entirely on ICL belief updates; avoiding these simplifications is emphatically backed up by our experimental results.
> >
> > Moreover, a large part of the success of our approach is in how good a model the LLM is for $p_{LLM}(y;\theta, x)$ compared to the efficacy of $p_{LLM}(\theta; x, y)$ (see previous response point) or the LLMs ability to choose questions directly without using a model-based approach.  By using a BED approach, we are exploiting the fact that the LLM is generally already good at answering specific questions.  For example, it can much more accurately answer the question “is this person male” if we tell it the person is Albert Einstein, than it can predict who the person is just from being told they are male.
> >
> > Regarding not updating the likelihood: again, this is a standard decision made by almost all Bayesian modeling approaches that stems directly from De Finetti’s theorem, and one that makes sense in the settings considered in this paper. Consider, for example, the game of 20 questions. Conditioned on the fact that the user is thinking of “cat”, the probability of them saying yes to “is it a mammal?” should not be changed by the fact they previously answered no to the question “is it a reptile?”. As we discuss in Appendix A.1, if this static likelihood model is not appropriate for a given use case, it easily can be modified to include the history, but this will clearly be inferior in practice for the problems we consider. We are happy to add an empirical ablation on this choice if you think it is important though.
> >
> > **W4 (compute)**. Our method is motivated by settings in which we are trying to minimize the amount of queries made to the user, such that we prefer to reduce the number of questions to the user, at the cost of computational overhead; best performance relative to compute time is thus not our aim.
> >
> > With that said, we agree that it is important that the computation time is not so large as to be prohibitive. Our compute time is dominated by LLM calls: we must generate M candidate questions, and N candidate hypotheses. We added the wallclock runtimes for 20 Questions for Qwen2.5-72B on the entire dataset animals in Table 2 in Appendix F. BED-LLM takes approximately 5x the wall-clock time of Naive. The Split and Entropy baselines involve the same number of LLM calls as BED-LLM, and so their runtime is similar to BED-LLM. Our hypothesis retention mechanism and efficient filtering have improved the wall clock time of LLM sampling significantly. Importantly, LLM calls at each stage are parallelizable, so runtime scales sublinearly with M and N.
> >
> > We also note that additional rounds of questioning are not always sufficient to get good performance out of direct usage of the LLM.  We have added a new Figure 8 to Appendix F that allows up to 50 questions to be asked for the 20 questions problem.  Here we see that the “Naive” baseline’s performance after 50 questions is still far below BED-LLM’s performance after 20 questions. This illustrates a key point: BED-LLM achieves better uncertainty reduction with fewer user interactions, than LLM prompting methods can achieve with substantially more user interactions. Extra computation spent on additional turns does not compensate for the absence of principled information-guided selection.

---

> ### Author Response · Authors · 2025-11-21
>
> **W5 (prompt-based methods)**. Thank you for this suggestion! We have now added ReAct-style “CoT” method as an additional baseline in the 20 Question experiment – see Figure 10, Appendix F. We will update the figures, tables and text in the main manuscript once the remaining experiments  have completed. As you can see in the updated manuscript, the approach is substantially outperformed by BED-LLM, and it also generally does noticeably worse than the baselines we had already considered.
>
> We also note that our “Implicit Maximization” baseline can be seen as a variant of the “Tree of Thoughts” method of Yao et al (2023), which is also noted by Kobalczyk et al. (2025), and so can be seen as a further prompt-based comparison on top of the Naive baseline already included. Specifically, it is a form of prompt-based planning, and can be viewed as a lightweight, inference-time ToT variant. We have made the link between Implicit Maximization and ToT explicit in Appendix G.
> We further emphasise that our Naive baseline already includes significant prompting to encourage good behaviour and elicit high quality questions – we should probably not have called it “Naive”!
>
> **W6a (experimental details)**. Thank you for highlighting this.  On reflection we agree that more details are needed for these baselines and so we have added a more detailed description of these baseline methods in Appendix G of the updated revision of the paper, and summarized below.  Additional details on general experimental setup are given in Appendices I and J.
>
> Naive:
> A direct prompting baseline where the model is asked to immediately generate the next question given the dialogue history, no candidate generation, and no belief modeling. This corresponds to the simplest and most common way users currently interact with LLM agents for question-asking tasks, and therefore serves as a natural “standard LLM prompting’’ baseline. After the LLM has generated its question, it waits for the user response and generates a new question based on the new history.
>
> Split:
> Essentially a version of our method in which we cast previous approaches, such as Kobalczyk et al 2025, which provides better performance than their own methods:
> 1. Uses the same candidate question generating method as BED-LLM,
> 2. Assumes a deterministic likelihood over hypotheses (i.e., each hypothesis deterministically predicts a single answer to each question), and
> 3. Evaluates the question on the sampled hypotheses, and selects the question that maximally balances the hypotheses, i.e., splits the current hypothesis set into subsets whose sizes are as close to equal as possible.
>
> **W6b (comparison methods)**. The “Naive” baseline is exactly a standard LLM prompting method with carefully tuned prompts; it is constructed to be as strong as possible while still following a standard agent strategy. It is based on the approach of Zhang et al (2024), which was found to surpass human performance on the 20 questions problem. As previously noted, we have since added a ReAct-style CoT method as an additional prompting based baseline as well. We find that our methods, including BED-LLM and Entropy, significantly outperform these.
>
> **W7 (ablations)**. We have ablated each of our design choices: The Entropy baseline corresponds to including only the marginal predictive entropy and not the expected likelihood entropy; the Implicit Max baseline corresponds to allowing the LLM to select the question rather than using EIG (while keeping the set of candidate questions constant); the “ICL Beliefs” ablation isolates the impact of the filtering method, while keeping EIG estimation the same; the Data–Estimation baseline looks at the impact of using a prior–likelihood pairing rather than a data–estimation pairing. Could you detail what specific ablations you feel are missing?
>
> **Q1 (approximations)**. As we discussed in our response to W3: We are not substituting some known ground truth $p(y|\theta, x)$ with an LLM proxy – the LLM is our generative model for an unknown underlying response model. The justification of our model setup, including this likelihood, is then in the overall strong empirical performance of BED-LLM, in particular its comprehensive outperformance of direct prompting strategies and the Data-Estimation approaches, which are the only known alternatives that can side-step specifying a likelihood model.

---

> > ### Author Response · Authors · 2025-11-21
> >
> > **Q2 (diverse behavior)**. Having “more nuanced and diverse behavior” is important to avoid the premature collapsing behaviour shown in Figure 5 in the appendix. BED is all about choosing designs that maximise our expected reduction in uncertainty, and if our model’s diversity collapses prematurely like in Figure 5, this also causes the uncertainty estimates to collapse prematurely and the EIG to no longer provide an appropriate objective. Need for diversity here is also doubly important because of the filtering procedure: it is acting like the proposal for a rejection sampler, which means we ideally want to be at least as diverse as the distribution being targeted by the rejection sampling.
> >
> > **Q3 (updating the likelihood model)**. See W3; this is a standard Bayesian assumption that fits our settings: In Bayesian inference, we typically use a static likelihood model to update our prior to obtain a posterior. As we discuss in Appendix A.1, this assumption can easily be relaxed; however in the settings explored in this paper there is no clear reason to believe that the likelihood would change based on interaction history, while including in context for the likelihood could induce spurious correlations with previous answers.
> >
> > **Q4 (allowable entities)**. To construct our estimator, we do use Monte Carlo sampling from the space of hypotheses to approximate expectations, with the number of samples inevitably finite. Such sample-based approximations allow us to obtain valid estimators of the true expectation, even when the space is unbounded. Conversely, several other methods explicitly assume a pre-fixed restricted set of candidates.  In particular, these previous approaches have generally assumed the same fixed set of candidates throughout the sequential questioning process, restricting their use to cases where the size of $\theta$ is small, known, and enumerable. By contrast, new $\theta$ samples are generated in BED-LLM at each step conditional on the previous answers, allows us to search through far larger spaces and find things the LLM would struggle to even envisage as the possible true $\theta$ at the start of the questioning process.
> >
> > > Is the key difference that prior work assumes  is explicitly given, while your approach allows  to be generated by the LLM?
> >
> > No, there are many differences to previous work and we would not say that it is the key difference either, though it is indeed an important one as it allows us to work with much more general theta spaces; most previous methods are not applicable to the preference elicitation task as $\theta$ is a free-form paragraph of text that we cannot enumerate over.  One of the most important differences is that we are estimating the expected information gain rather than just predictive entropy (see Section 3.2), which our experimental results show gives a significant improvement in performance. Other important differences include how our filtering strategy is set up (with the poor performance of the ICL beliefs approach showing is critical to get right), and providing substantial insights and formalisation of how to best apply BED in the LLM setting (cf Section 4, noting that previous papers never actually considered exactly how the model used by BED should be derived from the LLM and the nuances involved in this). We put substantial thought into how we should derive our probabilistic model from the LLM, which is key to the success of our approach.
> >
> > **Q5 (experimental details)**.
> > > Could you provide details on how the "Naive'' and "Split'' baselines are implemented?
> >
> > Please see W6 and the new sections detailing this in the Appendix G.
> >
> > > Additionally, how does the number of possible answers (20 vs. 500) affect performance?
> >
> > We interpret this as if the Reviewer meant the “number of turns”; if the Reviewer meant something different, please clarify. We added the results for 50 question-answer turns for Qwen2.5-72B in Appendix F of the updated revision of the paper. For turns t>20, the success rate growth slows down for many method+dataset combinations, and the BED-LLM remains the best performing method across the board.  Note that the performance of BED-LLM after 20 questions is generally still better than most baselines have achieved after 50, reinforcing how strong its performance is.
> >
> > **References**:
> > Kobalczyk, K., Astorga, N., Liu, T., and van der Schaar, M.. Active task
> > disambiguation with llms. In International Conference on Learning Representations, 2025.
> > Liu, N.F., Lin, K., Hewitt, J., Paranjape, A., Bevilacqua, M., Petroni, F., and
> > Liang, P.. Lost in the middle: How language models use long contexts. Transactions of the
> > Association for Computational Linguistics, 12, 2024.
> > Yao, S., Yu, D. Zhao, J., Shafran, I., Griffiths, T., Cao, Y., and Narasimhan, K..
> > Tree of thoughts: Deliberate problem solving with large language models. In Advances in Neural Information Processing Systems, 2023.

---

### Official Review · Reviewer_Ah5k · 2025-11-01

**Soundness:** 3
**Presentation:** 2
**Contribution:** 2
**Rating:** 6
**Confidence:** 4

**Summary:**

The authors propose BED-LLM, a method designed to improve the information gathering ability of large language models (LLMs). The main idea of the paper is to select questions that maximize the expected information gain (EIG), enabling the model to acquire relevant information more efficiently than existing approaches. The method requires no parameter updates or fine-tuning and can be easily used with any pretrained LLMs. Through experiments on the 20 Questions game and film preference elicitation tasks, the authors demonstrate that maximizing EIG significantly enhances an LLM’s ability to ask informative and discriminative questions. These results indicate that EIG is an effective metric for improving information gathering in LLMs.

**Strengths:**

-	BED-LLM goes beyond simply identifying questions that elicit new information. It also evaluates whether each question is contextually coherent and whether it contributes to determining the final answer. By incorporating these considerations, the method effectively avoids asking irrelevant or redundant questions, guiding the model toward more meaningful and goal-oriented information gathering.

-	The method is model-agnostic. It can be used with any existing LLMs without any parameter updating or fine-tuning, which greatly enhances its practicality and generalizability.

-	The authors decompose BED-LLM into four distinct components (Entropy, Data-Estimation, ICL Beliefs, Implicit Maximization)  and perform ablation experiments to isolate the contribution of each method. This experimental design enables a quantitative assessment of how each component contributes to the model’s information-gathering performance, offering empirical evidence of the method’s internal effectiveness.

**Weaknesses:**

-	The experimental evaluation is relatively limited in scope. The 20 Questions setting, while well-structured, represents a simplified and highly constrained environment that may not adequately capture the ambiguity and uncertainty inherent in real-world information-gathering tasks. Moreover, the chosen topics—animals, celebrities, and things—are narrow in domain and may not generalize well to more complex or abstract scenarios. This raises questions about the method’s robustness and applicability across diverse contexts.

-	The paper does not clearly define the termination criterion for the questioning process. In the 20 Questions experiment, the interaction terminates either when the model exhausts all twenty turns or when only a single hypothesis remains and the model correctly identifies it.
In contrast, in the film preference elicitation setup, the model proposes a list of films at each turn, which is then evaluated against the user’s preferences, but the stopping condition for this process is not clearly specified. As a result, it remains unclear how and when BED-LLM determines that sufficient information has been gathered to conclude the interaction.

**Questions:**

The BED-LLM approach appears to be computationally heavy, as it generates multiple candidate questions, estimates their expected information gain (EIG), and updates a belief distribution at each turn. It would be helpful to understand how the computational cost compares to baseline methods, such as Naive or Split, in terms of inference-time overhead.

---

> ### Author Response · Authors · 2025-11-21
>
> Thank you for the helpful review and feedback! We hope the responses below and the revised manuscript address your concerns and questions.
>
> **W1 (Experimental evaluation).** While relatively structured, the 20 Questions setting is by far the most commonly used benchmark for question asking in the literature, and is considered in most related works (many of which use the same “things” and “celebrities” datasets). We would argue that they are not particularly narrow in scope – the LLM does not have a shortlist of items to choose from, and (for example) the “things” dataset includes items such as “invisibility cloak”, “aristocrat” and "connoisseur". Moreover, it is representative and captures the fundamental structure of many real-world settings where the goal is to reduce uncertainty quickly about a discrete, unknown entity, under a limited question or testing budget, such as medical diagnosis.
> Despite its apparent simplicity, direct LLM prompting consistently fails on 20Q because the task stresses long-horizon belief tracking, consistency with past answers, and information-sensitive question formulation. These are precisely the challenges our method is designed to address.
>
> We complement this dataset with the highly unstructured movie preference dataset. The underlying profiles are expressed in free-form natural language, and are highly variable, containing multiple overlapping facets of a user’s movie preferences: there is a lot of variation in how a user could express their movie preferences. We believe that this noisy, unstructured setting much better mirrors real-world preference elicitation and ambiguity than the more structured preference elicitation settings that have been explored recently (e.g., by Handa et al., 2024). This setting is representative of more real-world “fuzzy” preference elicitation tasks, such as learning general preferences to be prepared for future queries, or adapting communication style to meet a user’s tastes, where the goal is to identify a latent persona from ambiguous signals.
> The experiments show BED-LLM’s success on complementary ends of the spectrum of information gathering that many real agents face. 20 Questions tests whether BED-LLM can effectively reduce hypotheses over discrete, answerable targets, while the preference elicitation setting tests whether the same method is robust for noisy, unstructured persona targets.
>
> **W2 (termination criterion).** We do not use a termination criterion for the film recommendation problem because there are only 5 rounds of questioning and early termination is not generally needed (as we never expect to have all possible knowledge about a user’s preferences). The termination criterion for 20 Questions is clearly defined as you explain yourself, and is an example of terminating based on determining sufficient information has been gathered (as this task is only done once we have identified the singular true answer).  If desired, one could easily add additional or alternative termination criteria based on more general criteria on how much information has been gathered (e.g. looking at entropy on theta given the history), but this was not needed for this task. We are happy to add further discussion on this if you think it is important, but we emphasise that BED-LLM is in no way restricted in this regard so we do not feel it should be viewed as a weakness of the paper.
>
> **Q1 (compute).** Our method is motivated by settings in which LLM computations are much “cheaper” than asking questions to the user (e.g., preference elicitation). Hence, we would much prefer to reduce the number of questions to the user, at the cost of computational overhead. With that said, compute time is dominated by LLM calls: we must generate M candidate questions, and N candidate hypotheses. We added the wallclock runtimes for 20 Questions for Qwen2.5-72B on the entire dataset animals in Table 2 in Appendix F. BED-LLM takes approximately 5x the wall-clock time of Naive. The Split and Entropy baselines involve the same number of LLM calls as BED-LLM, and so their runtime is similar to BED-LLM.
> We note that additional rounds of questioning using the LLM directly are not always sufficient to obtain good performance.  We have added a new Figure 8 to Appendix F that allows up to 50 questions to be asked for the 20 questions problem.  Here we see that the “Naive” baseline’s performance after 50 questions is still far below BED-LLM’s performance after 20 questions. This illustrates a key point: BED-LLM achieves better uncertainty reduction with fewer user interactions, than LLM prompting methods can achieve with substantially more user interactions. Extra computation spent on additional turns does not compensate for the absence of principled information-guided selection.
>
> **References**
>
> Handa, K., Gal, T., Pavlick, E., Goodman, N., Andreas, J. Tamkin, A.,  and
> Li, B.Z. Bayesian preference elicitation with language models. arXiv preprint arXiv:2403.05534, 2024.

---

### Official Review · Reviewer_T2zx · 2025-11-03

**Soundness:** 3
**Presentation:** 3
**Contribution:** 2
**Rating:** 4
**Confidence:** 4

**Summary:**

This paper presents BED-LLM, a framework that integrates BED principles into LLMs to enable adaptive and information-efficient query generation. The core idea is to model belief over hypotheses and select the next query by maximizing an approximate EIG, implemented through an LLM-mediated sampling and likelihood estimation process. The formulation adapts Bayesian inference perspectives into a framework for LLM-driven reasoning. The experiments, however, are limited to small, synthetic tasks and do not provide sufficient evidence that the proposed method yields general improvements in interactive reasoning or adaptive planning. The contribution is conceptually promising and well-articulated, but the empirical validation is too narrow to establish practical impact.

**Strengths:**

- Presents a clear and well-motivated connection between Bayesian experimental design and LLM-based information gathering.
- Offers a unified probabilistic framing that clarifies the relationship between prompting, belief updates, and adaptive query selection.
- Demonstrates measurable gains over naïve baselines on synthetic tasks, suggesting that EIG-guided prompting can improve efficiency in principle.

**Weaknesses:**

- Limited scope of evaluation. Although the paper is positioned as a general framework for intelligent and adaptive information gathering with LLMs, the experiments are confined to small, toy-scale domains (20-Questions and synthetic movie recommendation). These tasks are useful for illustration but are quite far from the complex, noisy, multi-step reasoning settings the paper emphasizes introduction emphasizes. As a result, the evaluation does not convincingly demonstrate that BED-LLM generalizes beyond simple controlled environments.
- Missing planning and reasoning baselines. The paper compares primarily against naïve or ablated versions of its own approach, omitting stronger baselines such as reinforcement learning, Monte Carlo tree search, or other uncertainty-driven planning methods that have been explored for adaptive query generation in LLMs. Without these comparisons, it is difficult to attribute observed gains to the proposed Bayesian-experimental-design formulation rather than to prompt structure or candidate filtering.
- Lack of validation for expected information gain (EIG) approximations. The core methodological claim is that BED-LLM uses approximations of expected information gain to guide question selection. However, there is no empirical or analytical evaluation of the accuracy or stability of these approximations. The paper reports only downstream success metrics, leaving unclear whether the observed performance improvements arise from better EIG estimation or unrelated modeling choices. A more direct analysis, for e.g., comparing true vs. approximated EIG, or assessing consistency under varying priors, would greatly strengthen the argument.

**Questions:**

See above weaknesses.

---

> ### Author Response · Authors · 2025-11-21
>
> Thank you for your questions and feedback! We hope our responses below and additions to the paper address your concerns.
>
> **W1 (scope of evaluation).** Our reasons for using the 20 Questions setting are twofold. Firstly, it is by far the most commonly used testbed for question-asking in recent LLM literature. Second, while it is a “toy” experiment, it is something that LLMs cannot generally handle by themselves.  Moreover, it is representative and captures the fundamental structure of many real-world settings where the goal is to reduce uncertainty about a discrete, unknown entity, under a limited question or testing budget, such as medical diagnosis. It also allows us to measure uncertainty reduction cleanly and to validate the belief-update loop under controlled conditions.
>
> Conversely, the synthetic movie recommendation setting is a noisy, complex setting. The underlying profiles are expressed in free-form natural language, and are highly variable, containing multiple overlapping facets of a user’s movie preferences: there is a lot of variation in how a user could express their movie preferences. We believe that this noisy, unstructured setting much better mirrors the challenges of real-world preference elicitation than the more structured preference elicitation settings that have been explored recently (e.g., by Handa et al., 2024). Our setting is representative of more real world “fuzzy” preference elicitation tasks - such as learning general preferences to be prepared for future queries, or adapting communication style to meet user’s tastes - where the goal is to identify a latent persona from ambiguous signals.
>
> The two domains span complementary ends of the spectrum of information gathering that many real agents face. 20 Questions tests whether BED-LLM can effectively reduce hypotheses over discrete, answerable targets, while the preference elicitation setting tests whether the same method is robust for noisy, unstructured persona targets.
>
> **W2a (planning/reasoning baselines).** In this paper, we focus on the development of an inference-time method, i.e., taking an existing LLM and deploying the best possible information-gathering strategy. As such, we do not compare with RL-based approaches that require fine-tuning and additional data to be available for this (although, we note that our test-time approach could be applied in conjunction with RL-finetuned models).
>
> While MCTS-type methods can be used at inference-time, full explicit MCTS planning is computationally prohibitive, generally requiring orders of magnitude more compute to be successful than we are using for BED-LLM. More implicit planning approaches can be used, but this is exactly what our “Implicit Maximization” baseline represents: it can be seen as a variant of the “Tree of Thoughts” method of Yao et al. (2023).
>
> The “Split” method serves as a strong baseline and represents what was previously generally considered state-of-the-art in the area,  our implementation is a steel-manned version of Kobalczyk et al. (2025) that gives better performance than their own implementation. The “Entropy” baseline improves on this slightly by making less restrictive assumptions on the deterministic likelihood: it already has better performance on 20 questions than previous methods in the literature have achieved.
>
> Based on the reviewer’s feedback, we have, though, added an additional strong baseline that reflects another possible planning-based approach: a chain-of-thought-based method, where the model is allowed to perform CoT reasoning before generating a question. This is in contrast to the “Naive” baseline, where the model outputs a question immediately. We’ve included these updated results in Appendix F (Figure 10). The new method performs comparably to the LLM-implicit baseline but remains well below BED-LLM, Entropy, and Split.

---

> ### Author Response · Authors · 2025-11-21
>
> **W2b (“It is difficult to attribute observed gains to the proposed BED formulation”).** We strongly refute the suggestion that our evaluations do not confirm the gains originate from our BED formulation. First, we have run various ablations (Data-Estimation, ICL beliefs, Entropy, and Implicit maximisation) to confirm that each component of the BED-LLM approach is essential in achieving good performance, rather than the gains coming from low-level algorithmic decisions. Second, we have carefully designed our experiments to make sure that the performance of BED-LLM is *not* due to better prompting or filtering: unless otherwise stated, we use the same prompts and filtering for all comparison methods. For example, the prompts, candidate questions and (for Split and Entropy) the candidate hypothesis generation/filtering methods in the “LLM implicit”, “Split”, and “Entropy” baselines are identical to in BED-LLM (which is why Split is a steel-manned version of Kobalczyk et al. (2025) whose original version did not use filtering); the only difference is how the final question is selected. Conversely, the “ICL Beliefs” ablation isolates the impact of the filtering method, while keeping EIG estimation the same.
>
> **W3 (EIG approximation).** We again strongly disagree that the observed performance improvements can come from unrelated modelling choices due to the various ablations we have run and careful experiment constructions.  The Rao-Blackwellized estimator that we use is known to be consistent under increasing number of $\theta$ samples (see Rainforth et al, 2018) and it is already well known that accurate EIG estimation is not always needed for effective BED provided the ranking between designs is conserved (see e.g. Foster et al, 2021, where the simple PCE estimator was found to often match or outperform more advanced estimators in terms of final design quality). However, to alleviate any remaining concerns on this, we have added results showing how the estimated EIG varies as we increase the number of candidate hypotheses (see Appendix F).
>
> Concretely, we sample a large pool of hypotheses $\Theta = \theta_1, \ldots, \theta_K$ from our constructed prior $p_f(\theta ; h_t)$. We then compute EIG estimates of the candidate questions using increasing independent subsets of these hypotheses (e.g. size 5, 10, 20, …). We plot the resulting EIG values as a function of subset size with error bars over several runs, and demonstrate that the estimated EIG stabilizes rapidly, with diminishing changes beyond 10-15 hypotheses. This indicates that the approximation is numerically stable and downstream performance gains are not due to estimation noise.
>
> **General (“Does not provide sufficient evidence of general improvements in interactive reasoning or active planning”).** We respectfully disagree with this point. In the 20 Questions setting, we show dramatic, consistent improvement over existing approaches (between 5 and 25 percentage points higher than the next best method and typically at least 30 percentage points better than direct usage of the LLM, with an **8x** improvement with Llama-3.1-8B). Further, we show consistent improvement in a noisy, realistic setting where many existing approaches are not even applicable and that our approach is robust to model mismatch (Figure 1).
>
> **References**
>
> Foster, A., Ivanova, D.R., Malik, I., and Rainforth, T. Deep Adaptive Design: Amortizing Sequential Bayesian Experimental Design. In Proceedings of the 38th International Conference on Machine Learning, 2021.
>
> Handa, K., Gal, T., Pavlick, E., Goodman, N., Andreas, J. Tamkin, A.,  and Li, B.Z. Bayesian preference elicitation with language models. arXiv preprint arXiv:2403.05534, 2024.
>
> Kobalczyk, K., Astorga, N., Liu, T., and van der Schaar, M.. Active task disambiguation with llms. In International Conference on Learning Representations, 2025.
>
> Rainforth, T., Cornish, R., Yang, H., Warrington, A., and Wood, F. On Nesting Monte Carlo Estimators. In Proceedings of the 35th International Conference on Machine Learning, 2018.
>
> Yao, S., Yu, D. Zhao, J., Shafran, I., Griffiths, T., Cao, Y., and Narasimhan, K. Tree of thoughts: Deliberate problem solving with large language models. In Advances in Neural Information Processing Systems, 2023.

---

### Official Review · Reviewer_Jn86 · 2025-11-07

**Soundness:** 3
**Presentation:** 3
**Contribution:** 3
**Rating:** 8
**Confidence:** 3

**Summary:**

This paper presents BED‑LLM. It treats multi‑turn question asking as sequential Bayesian experimental design. At each step, the model makes several candidate multiple‑choice questions. It then picks the one with the highest expected information gain (EIG) and asks it. After the answer, it updates a simple belief over possible hypotheses by sampling, filtering for consistency and renormalizing. The method uses a prior–likelihood setup and a low‑variance EIG estimator based on the model’s logits. It avoids the common mistake of using only predictive entropy. The authors show gains on 20 Questions and movie‑preference tasks, including when the questioner and answerer are different models.

**Strengths:**

- Clear problem formalization and principled objective. The paper cleanly frames multi‑turn question selection as sequential Bayesian experimental design (BED) and optimizes incremental expected information gain (EIG) in a concrete loop: generate candidate questions -> estimate EIG  -> select the best question -> update the belief state.
- Rigorous treatment of uncertainty (beyond predictive entropy). The authors explicitly show that predictive entropy ≠ EIG and that dropping the expected likelihood entropy term selects ambiguous or irrelevant questions. They provide an intuitive example and ablations (“Entropy” baseline) demonstrating real performance gaps.
- Sensible modeling choice. It uses a prior–likelihood factorization that puts uncertainty in the answer space y (small and enumerable) rather than the often much bigger hypothesis space $\theta$. Practically, questions are written as multiple‑choice so probabilities and entropies are reliable.
- Belief update that fixes in‑context issues. Instead of relying on raw in‑context updates (which can be inconsistent and overconfident), it samples hypotheses, filters them against the history, retains consistent ones across turns, and reweights uniformly. Removing this step hurts performance in ablations, which shows it matters.

**Weaknesses:**

- No compute time accounting. Each turn can be heavy: sample at least N=15 hypotheses (with up to three generate‑and‑filter rounds), score M=15 candidate questions, etc. The paper doesn’t report tokens, wall‑clock, or cost curves to show the overhead vs. baselines.
- Sensitive to API log‑probabilities. The EIG estimator and the filtering step assume access to stable token‑level probabilities. Some APIs limit this.
- The algorithm optimizes the next question’s EIG only. The paper itself notes that sequential BED can be suboptimal and points to policy‑based alternatives, but it doesn’t compare against them here.
- Depends on multiple‑choice answers. The method works by keeping the answer space small and enumerable, so it doesn’t directly handle open‑ended text or continuous answers. This is by design to make probabilities and entropies reliable, but it narrows where the approach applies.

**Questions:**

- Why threshold 0.2, and why uniform reweighting? How sensitive are results to these, and what happens with temperature scaling?
- For free‑text or continuous y, what EIG approximations or discretizations are viable?

---

> ### Author Response · Authors · 2025-11-21
>
> Thank you for your positive review and appreciation of our clear formalization, principled objective, and rigorous treatment of uncertainty. We hope our response and additions to the paper will address your concerns.
>
> **W1 (compute time).** Thank you for highlighting this. Our method is motivated by the settings in which LLM computation is far cheaper than asking additional questions to the user (e.g. preference elicitation or clinical triage) . Hence, we would much prefer to reduce the burden on the user, at the cost of computational overhead.
>
> Having said that, we agree that reporting compute cost is valuable. We have added wall clock times (Table 2, in the Appendix); on the Animals dataset. We note that most of the LLM calls involved, including candidate question scoring, likelihood evaluations for answer choices, and hypothesis filtering, can be parallelized, hence the latency generally wouldn’t scale with M or K for optimized deployments in practice.
>
> BED-LLM takes approximately 5x the wall-clock time of Naive. The Split and Entropy baselines involve the same number of LLM calls as BED-LLM, and so their runtime is similar to BED-LLM.
>
> **W2 (API calls).** If the API doesn’t return log-probabilities, it is still possible to use a sample-based estimator like those employed in Hu et al. (2024) and Kobalczyk et al. (2025), but it will not be Rao-Blackwellized and thus will have provably higher variance. If possible, one should use log-probabilities, which would be available in all local models and some APIs. Because we are working with a small space of possible responses, access to the top-K log-probabilities is generally sufficient (cf the GPT results where we only have access to these). If log-probabilities are not available then we can fall-back to the sampling-based EIG estimator and the framework would still work.
>
> **W3 (policy-based alternatives).** One could indeed look to extend our approach to the policy-based BED setting.  However, training these generally requires huge numbers of rollouts of the full experiment process (typically 100,000+ in more classical BED settings) which would make them extraordinarily expensive to train in the LLM setting. Similarly, while non-myopic inference time BED approaches also exist, they will generally be orders of magnitude more expensive to implement than BED-LLM.
>
> **W4 (multiple choice).** Yes, our designs (questions) are planned such that they can elicit answers which can form a categorical distribution. This is an assumption/paradigm that we state explicitly in Section 4: “we should generally favor the prior–likelihood formulation if $\theta$ is more complex and the data–estimation formulation if $y$ is more complex.” Extension to problems with more open-ended responses is a natural direction for future work; see also Q2.
>
> **Q1 (threshold/uniform reweighting).** The relatively low threshold of 0.2 corresponds to rejecting samples only when we are reasonably confident it is an incompatible sample and was found to give a good balance between preventing overzealous filtering (where valid samples are removed) and catching the vast majority of errors. Note here that there can be ambiguity in the true response: for example, the question “Is Leonardo da Vinci a writer?” may be answered positively or negatively, but there is sufficient probability on a positive answer to not remove $\theta$ = Leonardo da Vinci from our set.
>
> We use uniform reweighting to reflect an uninformative prior, which we use because directly using the LLM’s distribution over hypotheses leads to a highly peaked distribution, placing very high probability mass on the top few answers in a way that we do not think is likely to reflect our actual beliefs (see for example Figure 5 where the LLM is very overconfident and this needs to be corrected). Temperature scaling with temperatures greater than one (along with co-sampling) increases the diversity in generated hypotheses, but we found the distribution still to be highly peaked, hence the uniform reweighting is still required.
>
> **Q2 (free text/continuous y).** Here one can generally use either sample-based estimators like PCE and NMC or variational-based approximations (see e.g. Rainforth et al (2024) for a recent review of these).
>
> For free-text y, it may also be helpful to utilise semantic clustering to group similar responses (akin to discrete semantic entropy in Farquhar et al (2024), and looking at the frequencies of each cluster. This would help to ensure we maintain a meaningful uncertainty estimate for such problems, as raw token entropy is unlikely to provide a meaningful signal here.
>
> **References**
>
> Farquhar, S., Kossen, J., Kuhn, L., and Gal, Y.. Detecting hallucinations in large language models using semantic entropy. Nature, 630:625–630, 2024.
>
> Rainforth, T., Foster, A., Ivanova, D.R., and Bickford Smith, F.. Modern Bayesian experimental design. Statistical Science, 39(1):100–114, 2024.

---

### Author Response · Authors · 2025-12-03
**Response to All Reviewers and AC**

We thank the reviewers for their careful and constructive feedback. Based on their comments, we have made a number of concrete additions to the paper:
1. **Computational cost**: We include wall-clock computational costs for all methods in Appendix F (Table 2), enabling a direct comparison under the same experimental setup.
2. **Additional baseline**: We add a ReAct-style CoT baseline to our existing baselines and ablations. We evaluate this on both 20 Questions and film preference elicitation, and find that BED-LLM remains consistently better in both domains (results in Appendix F). We also clarify the relationship between our “Implicit Maximization” ablation and prior search-based prompting approaches in Appendix G, in particular “Tree of Thoughts”, to better contextualize our performance gains and comparisons.
3. **Likelihood update ablation**: We add an ablation (Appendix F, Table 3) comparing the static likelihood we use in the model factorization for BED-LLM to a history-conditioned likelihood that is updated at each step, $p_{\text{LLM}}(y_{t + 1}; [\theta, x_{t + 1}, h_{t - 1}])$. This variant performs slightly worse than BED-LLM, supporting the factorization/design choice made in the main method and discussed in Appendix A.1.
4. **Validation of the EIG estimation**: To address questions about the accuracy/stability of our EIG estimation, we add a plot demonstrating the empirical convergence of our Rao-Blackwellized EIG estimator as the number of $\theta$ samples increases (Appendix F, Figure 9). We reiterate that the Rao-Blackwellization gives a provably lower-variance estimator than alternatives.
5. **Longer horizon interaction**: We add a plot of success rates on the 20 Questions game with 50 turns (vs. 20) and show that the significant gains of BED-LLM persist in the longer-horizon setting (Appendix F, Figure 8).
6. **Additional details and examples**: We have expanded the details of the different methods in Appendix G, and added more illustrative examples in Appendix H to make the subtler modelling discussions easier to follow.

---

### Meta-Review · Area_Chair_pdTx · 2026-01-06

**Summary:**

The main concerns raised by reviewers were:
1. Limited scope of evaluation (T2zx, Ah5k, Jn86). Reviewers expressed concerns that the datasets were too small to be informative.
2. Missing reasoning and prompt-based baselines (T2zx, 4sXe). Reviewers were concerned that the suite of baselines did not sufficiently cover the space of alternate approaches.
3. Missing compute time accounting (Jn86). Jn86 was concerned about the lack of results comparing compute time across the various methods.
4. Justification for simplifying assumptions (4sXe). 4sXe raised concerns about the validity of using an LLM to construct joint distributions over quantities of interest.

**Reviewer Concerns:**

Each of these concerns were addressed by the rebuttal.
1. The authors explained the role of 20 Questions as a common testbed for evaluating LLM capabilities. They also discussed how the movie preference dataset is a natural noisy complement.
2. The authors added a new ReAct-style chain of thought baseline.
3. The authors added wall clock times and clarified that the primary goal is to reduce the number of rounds of interaction.
4. The authors clarified that the simplifying assumptions follow directly from the principles of Bayesian statistics.

**Reviewer Scores:**

- T2zx is very likely to increase their score. The rebuttal clearly addresses concerns of limited evaluate scope and adds a new ReAct-style chain of thought baseline.
- Ah5k is about as likely as not to increase their score. Although the authors clarified the experimental scope in their rebuttal, the initial review was already positive.
- 4sXe is very likely to increase their score. The authors clarified several points about how the assumptions follow from Bayesian statistics and added a ReAct-style chain of thought baseline.
- Jn86 is very likely to keep their score. The initial review was already positive. The author rebuttal clarified the evaluation scope, added wall clock times, and explained how the method could be applied even when an API does not return log probabilities.

---

### Decision · Program_Chairs · 2026-01-26

Accept (Poster)